# Meiotic DNA break resection and recombination rely on chromatin remodeler Fun30

Pei-Ching Huang[1,2,7], Soogil Hong[3], Hasan F Alnaser [ID][4], Eleni P Mimitou[2,8], Keun P Kim [ID][3,5], Hajime Murakami [ID][2,4✉] & Scott Keeney [ID][1,2,6✉]

## Abstract

DNA double-strand breaks (DSBs) are nucleolytically processed to generate single-stranded DNA for homologous recombination. In *Saccharomyces cerevisiae* meiosis, this resection involves nicking by the Mre11–Rad50–Xrs2 complex (MRX), then exonucleolytic digestion by Exo1. Chromatin remodeling at meiotic DSBs is thought necessary for resection, but the remodeling enzyme was unknown. Here we show that the SWI/SNF-like ATPase Fun30 plays a major, nonredundant role in meiotic resection. A *fun30* mutation shortened resection tracts almost as severely as an *exo1-nd* (nuclease-dead) mutation, and resection was further shortened in a *fun30 exo1-nd* double mutant. Fun30 associates with chromatin in response to DSBs, and the constitutive positioning of nucleosomes governs resection endpoint locations in the absence of Fun30. We infer that Fun30 promotes both the MRX- and Exo1-dependent steps in resection, possibly by removing nucleosomes from broken chromatids. Moreover, the extremely short resection in *fun30 exo1-nd* double mutants is accompanied by compromised interhomolog recombination bias, leading to defects in recombination and chromosome segregation. Thus, this study also provides insight about the minimal resection lengths needed for robust recombination.

**Keywords** Exo1; Fun30; Meiosis; Recombination; Resection
**Subject Categories** Cell Cycle; DNA Replication, Recombination & Repair

## Introduction

During meiosis, recombination yields crossovers between homologous chromosomes, providing physical connections important for proper segregation in the first division (Hunter, 2015). Meiotic recombination is initiated by DSBs catalyzed by Spo11 in a topoisomerase-like reaction that leaves Spo11 covalently linked to the newly created 5′ DNA termini (Bergerat et al, 1997; Keeney et al, 1997; Neale et al, 2005) (Fig. 1A).

DSB ends are then resected in two steps (Mimitou and Symington, 2009) (Fig. 1A). Endonucleolytic cleavage of each Spo11-linked DNA strand by MRX/Sae2 provides entry sites for two exonucleases with opposite polarities: 3′-to-5′ Mre11 exonuclease activity (digesting toward Spo11) and 5′-to-3′ Exo1 exonuclease activity (digesting away from the DSB) (Cannavo and Cejka, 2014; Garcia et al, 2011; Mimitou et al, 2017; Neale et al, 2005; Zakharyevich et al, 2010). Consequently, Spo11 proteins at DSB ends are released with covalently linked oligonucleotides (Spo11 oligos) (Neale et al, 2005). The resulting 3′ single-stranded DNA (ssDNA) tails, which average ~800 nt in length, are bound by strand-exchange proteins Rad51 and Dmc1, setting recombination in motion (Brown and Bishop, 2014).

Most DSBs in *S. cerevisiae* occur within hotspots that usually span <200 bp and mostly correspond to the nucleosome-depleted regions (NDRs) at transcription promoters (Baudat and Nicolas, 1997; Ohta et al, 1994; Pan et al, 2011; Wu and Lichten, 1994). Because these NDRs are typically flanked by nucleosome arrays, the resection nucleases must traverse several nucleosomes' worth of DNA (Fig. 1A). However, while Exo1 is a processive enzyme in vitro with an average run length of ~6 kb on naked DNA (Myler and Finkelstein, 2017; Myler et al, 2016), it is strongly blocked by nucleosomes (Adkins et al, 2013). Moreover, MRX/Sae2 preferentially nicks near but not within nucleosomes in vitro (Wang et al, 2017) and nucleosomes impede MRX/Sae2 incision in vegetative cells (Gnugge et al, 2023). Therefore, the resection nucleases may require chromatin remodeling to be able to digest nucleosomal DNA in vivo (Fig. 1A). Multiple chromatin remodelers appear to play partially overlapping roles in the resection of DSBs in vegetative cells, including RSC, INO80, and Fun30 (Karl et al, 2021), but whether the same is true in meiosis has not yet been established.

We previously mapped resection endpoints genome-wide by digesting ssDNA with nuclease S1 followed by adaptor ligation and deep sequencing (S1-seq) (Mimitou and Keeney, 2018; Mimitou et al, 2017; Yamada et al, 2020). We found that population averages of resection endpoints are modestly enriched at the preferred positions of linkers between nucleosomes and that altered local chromatin structure (caused by the elimination of transcription factors Bas1 or Ino4) was associated with changes in resection

[1]Weill Cornell Graduate School of Medical Sciences, Cornell University, New York, NY 10021, USA. [2]Molecular Biology Program, Memorial Sloan Kettering Cancer Center, New York, NY 10065, USA. [3]Department of Life Science, Chung-Ang University, Seoul 06974, South Korea. [4]Chromosome and Cellular Dynamics Section, Institute of Medical Sciences, University of Aberdeen, Aberdeen AB25 2ZD, UK. [5]Research Center for Biomolecules and Biosystems, Chung-Ang University, Seoul 06974, South Korea. [6]Howard Hughes Medical Institute, Memorial Sloan Kettering Cancer Center, New York, NY 10065, USA. [7]Present address: Metagenomi, Emeryville, CA 94608, USA. [8]Present address: Immunai, 430 E 29th St, New York, NY 10016, USA. ✉E-mail: hajime.murakami1@abdn.ac.uk; s-keeney@ski.mskcc.org

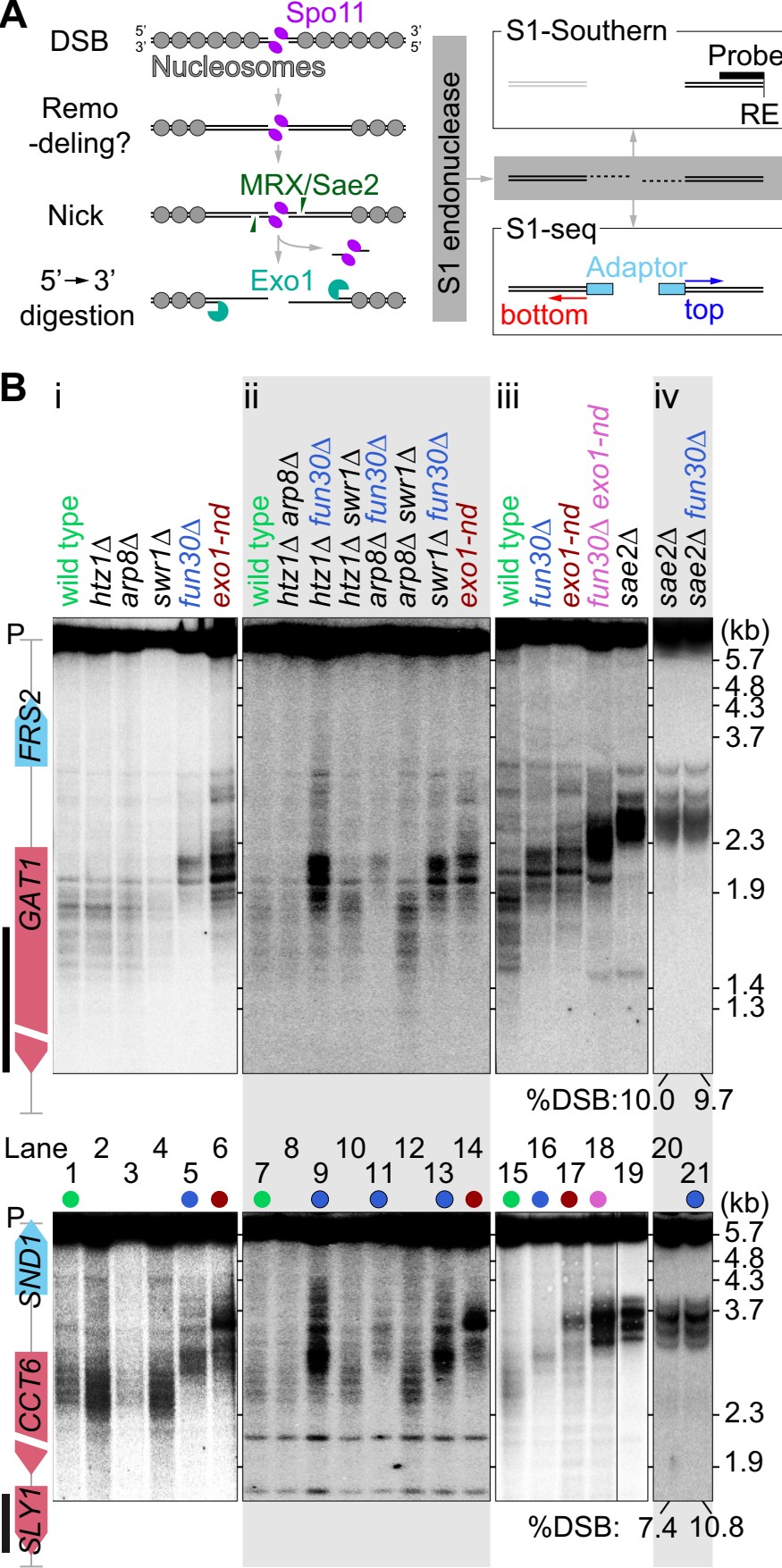

**Figure 1.  Testing chromatin remodeler mutants for shortened meiotic resection.**

(A) Left, an overview of meiotic DSB formation and resection within the context of local chromatin structure. Right, schematic of S1-Southern blotting and S1-seq methods. RE restriction enzyme. (B) Resection endpoint distributions detected by S1-Southern blotting at the *GAT1* (top) and *CCT6* (bottom) hotspots in the indicated mutants. All samples were collected at 4 h in meiosis. Vertical black lines to the left of the gene maps indicate probe positions. P parental length restriction fragments. Blot images are representative of n = 3 biological replicates except for *sae2Δ* and *sae2Δ fun30Δ* (n = 2 biological replicates). Source data are available online for this figure.

tracts (Mimitou et al, 2017). Moreover, computational simulations suggested that Exo1 digestion rates in vivo are comparable to rates in vitro on naked DNA (Mimitou et al, 2017). These findings suggested a scenario in which Exo1 runs on DNA duplexes that are effectively nucleosome-free, and then terminates digestion upon encountering the first intact nucleosome (Fig. 1A). This model reinforced the idea that nucleosomes may be actively removed or destabilized from broken chromatids before Exo1-mediated resection.

Here, we investigate the roles of chromatin remodelers in meiotic resection. We demonstrate that Fun30 is locally recruited in response to meiotic DSB formation and strongly influences the spatial patterns of both MRX/Sae2 nicking and further Exo1 processing. We further leverage the greatly shortened resection tracts in the *fun30 exo1-nd* double mutant to explore the functional importance of normal ssDNA tail lengths for recombination partner choice, crossover formation, and production of viable spores.

## Results

### Efficient resection requires Fun30

To identify chromatin remodelers with roles in meiotic resection, we screened candidate factors that have known effects on resection in vegetative cells. We tested mutations eliminating Fun30, a SWI/SNF-like ATPase that promotes resection for long distances (>5 kb away from the break site) (Chen et al, 2012; Costelloe et al, 2012; Eapen et al, 2012); Arp8, a nonessential subunit of the INO80 chromatin remodeler (Gospodinov et al, 2011; Morrison et al, 2004; Tsukuda et al, 2005); and Swr1, the ATPase subunit of the SWR-C complex (Papamichos-Chronakis et al, 2006; van Attikum et al, 2004). We also tested a mutant lacking histone variant H2A.Z (*htz1Δ*) because of the reported effects of H2A.Z on resection in vivo (Lademann et al, 2017; van Attikum et al, 2007) and on Exo1 digestion of nucleosomes in vitro (Adkins et al, 2013).

We examined resection at two strong DSB hotspots (*CCT6* and *GAT1*) using S1-Southern blots, in which genomic DNA from meiotic cultures was digested with nuclease S1 to remove ssDNA and then visualized by Southern blotting and indirect end labeling after agarose gel electrophoresis (Fig. 1A,B). Wild-type cells displayed a ladder of blunted DSB fragments that migrated considerably faster on the gel than the unresected DSBs seen in a *sae2Δ* mutant (Fig. 1B, compare lanes 1 and 15–19). The banding pattern in wild type reflects at least in part the effect of chromatin structure on the positions of resection endpoints (Mimitou et al, 2017). As previously shown (Mimitou et al, 2017), resection lengths are shorter in nuclease-dead *exo1-nd* mutants (*exo1-D173A*; (Tran et al, 2002)), so S1-digested DSB fragments migrated more slowly than in wild type (Fig. 1B, lanes 6, 14, 17).

The *fun30Δ* mutant similarly exhibited slower-migrating S1-treated fragments than wild type, with the magnitude of the change comparable to but distinct from that seen in *exo1-nd* (Fig. 1B, lanes 5 and 16). In contrast, none of the other mutations tested gave shortened resection lengths, either alone or in combination with one another, and none of them caused further changes in resection in a *fun30Δ* background (Fig. 1Bi,ii). We did not detect any difference in DSB distributions between *FUN30* and *fun30Δ* in the *sae2Δ* background (Fig. 1Biv), ruling out the possibility that *fun30Δ* affects resection endpoints only indirectly by altering DSB locations. We conclude that Fun30 is required to generate resection tracts of normal length.

DSB fragments in a *fun30Δ exo1-nd* double mutant migrated even more slowly than either the *fun30Δ* or *exo1-nd* single mutants (Fig. 1Biii). The resection seen in an *exo1-nd* background reflects the distribution of the MRX/Sae2 nicks that are the furthest from each DSB. Therefore, seeing shorter resection tracts in the *fun30Δ exo1-nd* double mutant compared to *exo1-nd* alone indicates that Fun30 also influences where MRX/Sae2 can act.

We note that human EXO1 protein with the equivalent substitution (D173A) retains weak nuclease activity, but at greatly reduced levels compared to wild-type (56-fold decreased on a blunt DNA end and 669-fold decreased for 5′-flap removal (Lee et al, 2002)). We therefore cannot exclude that some of the resection in the yeast *exo1-nd* mutant background is carried out by Exo1. However, we consider this unlikely to be a significant contribution because prior studies have documented highly similar resection defects in *exo1* null and *exo1-nd* mutants in both yeast and mice (Mimitou et al, 2017; Paiano et al, 2020; Yamada et al, 2020; Zakharyevich et al, 2010).

### Fun30 shapes the global resection landscape

To interrogate the genome-wide distribution of resection endpoints, we performed S1-seq on cells collected at 4 h after meiotic induction in wild type, *exo1-nd*, *fun30Δ*, and *fun30Δ exo1-nd*. Biological replicates correlated well (Appendix Fig. S1A), so they were averaged for further analyses. As previously shown (Mimitou et al, 2017), S1-seq reads at hotspots mapped to the top strand for resection moving rightward and to the bottom strand for leftward resection, and these S1-seq signals shifted closer to hotspot centers in *exo1-nd* (Fig. 2A). Consistent with the S1-Southern results, S1-seq signals were also closer to the hotspot center in *fun30Δ* and even closer in *fun30Δ exo1-nd* (Fig. 2A).

Global resection patterns were displayed by co-orienting and combining top- and bottom-strand reads around 3908 previously defined hotspots (Mohibullah and Keeney, 2017) and plotting the average relative to hotspot centers (Fig. 2B). Both single mutants had distributions shifted toward the hotspot midpoints, but with notable differences: *exo1-nd* showed substantially more signal than *fun30Δ* within 200 nt of hotspot midpoints, while *fun30Δ* showed a

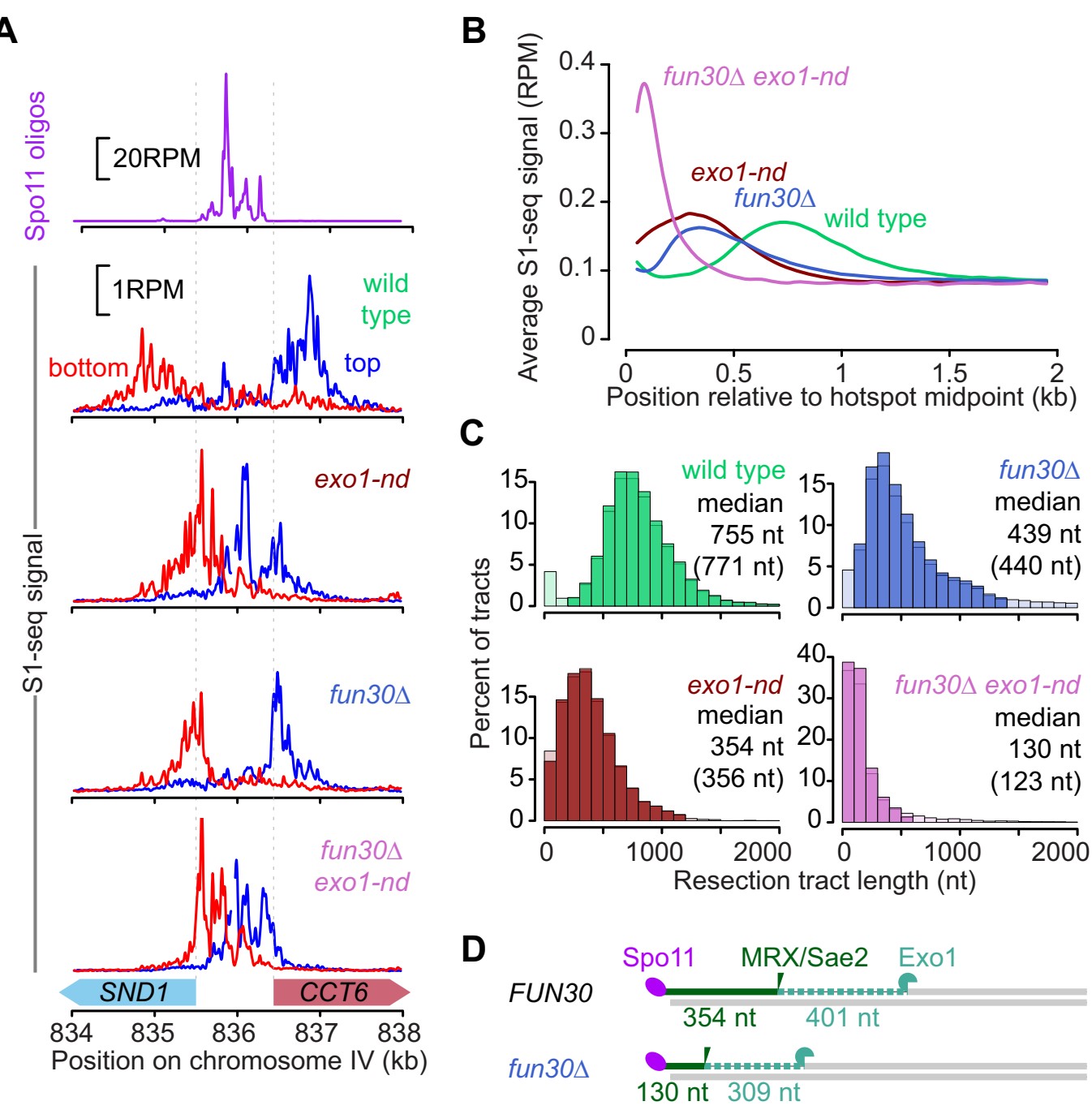

**Figure 2. The meiotic resection landscape in *fun30Δ* mutants.**

(A) S1-seq signals around the *CCT6* hotspot in reads per million mapped (RPM). Reads mapping to the top strand are shown in blue; bottom-strand reads are in red. All S1-seq data were the average of two biological replicates collected at 4 h in meiosis. Spo11-oligo data are from (Pan et al, 2011). (B) Genome average of S1-seq signal around 3908 hotspots. Bottom-strand reads were reoriented and combined with the top strand to calculate the average. Data were smoothed with a 100-bp Hann window. (C) Histograms of resection tract lengths calculated for "loner" hotspots that had no other hotspot within 3 kb (*n* = 405). Lighter colored bars indicate tracks that were omitted to calculate the censored median estimates shown in parentheses. Censoring had little effect, indicating that the measurements are not strongly influenced by outliers. (D) Schematic comparing *FUN30* and *fun30Δ* for the distance to the most distal MRX/Sae2 nicking positions (as measured in *exo1-nd* mutants) and the inferred lengths of Exo1 digestion.

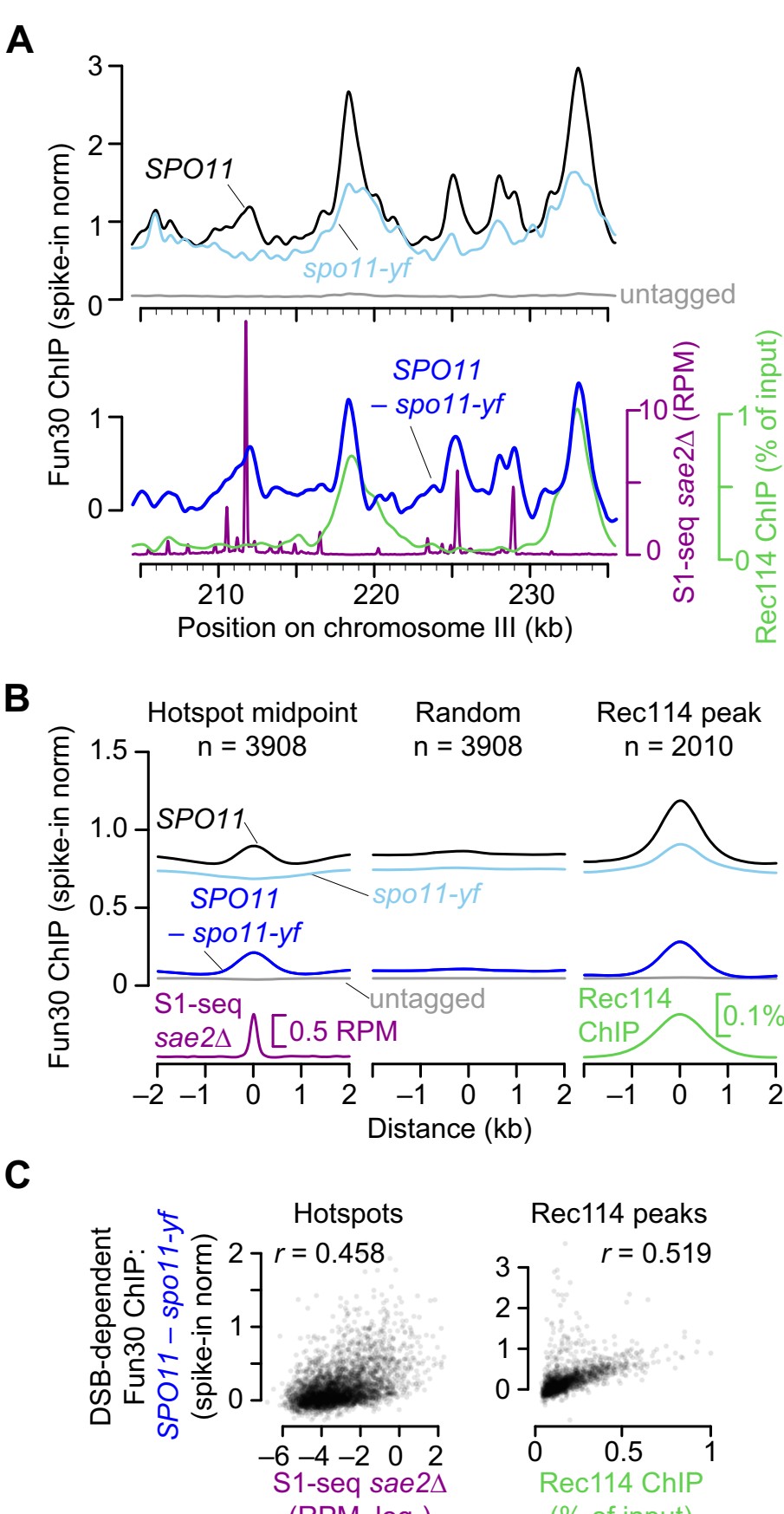

**Figure 3.    DSB-dependent recruitment of Fun30 to chromatin.**

(A–C) Fun30-myc ChIP-seq data were the average of 2 biological replicates collected at 4 h in meiosis. The DSB map (S1-seq in a *sae2Δ* mutant) is the same dataset shown in Appendix Fig. S1A,B as *sae2Δ* #2. The Rec114-myc ChIP-seq sample collected at 4 h in meiosis is from a previous study (Murakami and Keeney, 2014). Fun30 ChIP-seq datasets were normalized using a spike-in control. All data were smoothed using a 1 kb Parzen (triangular) sliding window. (A) Fun30 ChIP-seq signals across a representative region of chromosome III. The upper graph shows the normalized ChIP-seq coverage profiles for the wild type, the DSB-defective *spo11-yf* mutant, or a wild-type strain carrying untagged Fun30. The lower graph shows the DSB-dependent component of the Fun30 ChIP-seq signal, obtained by subtracting the normalized values from the *spo11-yf* strain from the values from the tagged wild-type strain. DSBs and Rec114 ChIP-seq profiles are shown for comparison. (B) Average Fun30 ChIP-seq signals around previously defined DSB hotspots (Mohibullah and Keeney, 2017) and Rec114 peaks (Murakami and Keeney, 2014). Profiles around a set of random genomic positions are shown for comparison. (C) Correlations (Pearson's *r*) between the DSB-dependent Fun30 ChIP-seq signals (summed in 1-kb windows) and both DSBs (left graph) or Rec114 ChIP-seq (right graph).

modestly higher frequency of longer resection tracts (~700–1200 nt) (Fig. 2B). These differences indicate that, in the absence of Fun30, the combined action of MRX/Sae2 and Exo1 resects nearly all DSBs at least 200 nt and occasionally resects almost as extensively as the longer tracts in wild-type. In contrast, MRX/Sae2 aided by Fun30 but without Exo1 nuclease often resects for only very short distances (<200 nt). The *fun30Δ exo1-nd* double mutant had even shorter resection tracts, with a modal endpoint within 100 nt of hotspot midpoints (Fig. 2B).

We estimated median resection tract lengths by focusing on 405 narrow (<400 bp wide) "loner" hotspots that had no other hotspots within 3 kb. This analysis gave median resection lengths of 755 nt in wild type and 354 nt in *exo1-nd* (Fig. 2C), comparable to previous results (Mimitou et al, 2017). Median resection was shortened to 439 nt in the *fun30Δ* mutant, and to 130 nt in the *fun30Δ exo1-nd* double mutant (Fig. 2C).

The difference between *exo1-nd* and *fun30Δ exo1-nd* reinforces the inference from S1-Southern data that the *fun30Δ* mutation alters MRX/Sae2 nicking positions. Importantly, however, the S1-seq data also suggest that Fun30 promotes Exo1-dependent resection as well. If we assume that the presence or absence of Exo1 nuclease activity does not influence MRX/Sae2 nicking positions, then we can estimate the contribution of Exo1 to total resection as the difference between *EXO1+* and *exo1-nd* backgrounds. By this analysis, Exo1 resects further in the presence of Fun30 (median$^{\text{wild type}}$ – median$^{exo1-nd}$ = 401 nt) than when Fun30 is missing (median$^{fun30Δ}$ – median$^{fun30Δ\ exo1-nd}$ = 309 nt) (Fig. 2D). We infer that Fun30 likely affects both resection steps.

S1-seq in a *sae2Δ* background, which maps DSB locations (Mimitou and Keeney, 2018; Mimitou et al, 2017), confirmed that the *fun30Δ* mutation had little if any effect on either the frequency or distribution of DSBs within hotspots (Appendix Fig. S1B,C). Thus, Fun30 does not affect where Spo11 cleaves but instead influences resection per se. S1-seq further confirmed that Swr1 is dispensable for resection genome-wide (Appendix Fig. S1D–F). We also performed S1-seq on cells lacking Rad9, which has an inhibitory effect on resection in vegetatively growing yeast (Chen et al, 2012), but detected little or no difference from wild type (Appendix Fig. S1D–F).

## Fun30 is recruited to chromatin in response to meiotic DSBs

To test if Fun30 is recruited to sites of meiotic DSBs, we performed chromatin immunoprecipitation followed by deep sequencing (ChIP-seq) of Fun30-myc in *SPO11* and *spo11-yf* strains (Y135F; catalytically inactive mutant). The methods we used would not

generate sequencing reads from the ssDNA left after resection. Moreover, we reasoned that any direct role of Fun30 in remodeling the chromatin of broken chromatids should occur prior to the onset of resection and that resection might remove proteins that were bound to the DNA at the time of nuclease action. Therefore, we conducted these experiments in a *sae2Δ* background to prevent resection. To quantitatively compare ChIP-seq signals between datasets, we used *Saccharomyces mikatae* cells expressing myc-tagged Rec114 as a spike-in control.

Both *SPO11* and *spo11-yf* strains showed pan-genomic Fun30 ChIP-seq signals that were on average more than 16-fold above an untagged control (Figs. 3A and EV1A). In both strains, Fun30 was enriched in genomic regions where its preferential binding has been previously reported, including tRNA genes, replication origins (ARS), and centromeres (Durand-Dubief et al, 2012; Neves-Costa et al, 2009) (Fig. EV1B,C). The DSB-independent component of this basal chromatin association presumably reflects the constitutive roles of Fun30 in chromatin modulation throughout the genome.

Importantly, there was also pronounced DSB-stimulated Fun30 enrichment at hotspots and other genomic regions. To investigate this DSB-dependent component of Fun30 binding, we plotted the difference in calibrated ChIP-seq signal between the *SPO11* and *spo11-yf* datasets (Fig. 3A, bottom). We observed peaks that coincided with DSB sites (i.e., peaks in the *sae2Δ* S1-seq map), but also peaks that lined up with peaks in ChIP-seq maps for the Spo11-accessory factor Rec114. The Rec114 peaks are thought to be preferred sites of assembly of DSB-promoting machinery along chromosome axes (Murakami and Keeney, 2014; Panizza et al, 2011).

The average Fun30 ChIP-seq signal around DSB hotspots showed a peak in *SPO11* but a modest depression in *spo11-yf*, so the averaged difference map had a broad peak extending ~500 bp to each side of hotspot centers (Fig. 3B). No such enrichment was observed for a random sample of genomic loci (Figs. 3B and EV1C). Furthermore, the local DSB-dependent ChIP-seq signal was correlated with hotspot strength (Fig. 3C), supporting the conclusion that Fun30 is recruited in cis to sites where DSBs have occurred.

Unlike around hotspots, the Fun30 ChIP-seq signal was already enriched in the absence of DSBs (*spo11-yf*) at preferred Rec114 binding sites (*n* = 2010, (Murakami and Keeney, 2014)), and became further enriched in the *SPO11* strain (Figs. 3B and EV1C). The DSB-dependent component of the Fun30 ChIP-seq signal was correlated with the Rec114 ChIP-seq signal (Fig. 3C). Similarly, the other genomic features that showed basal enrichment of Fun30 (tRNA genes, ARSs, centromeres) also showed further

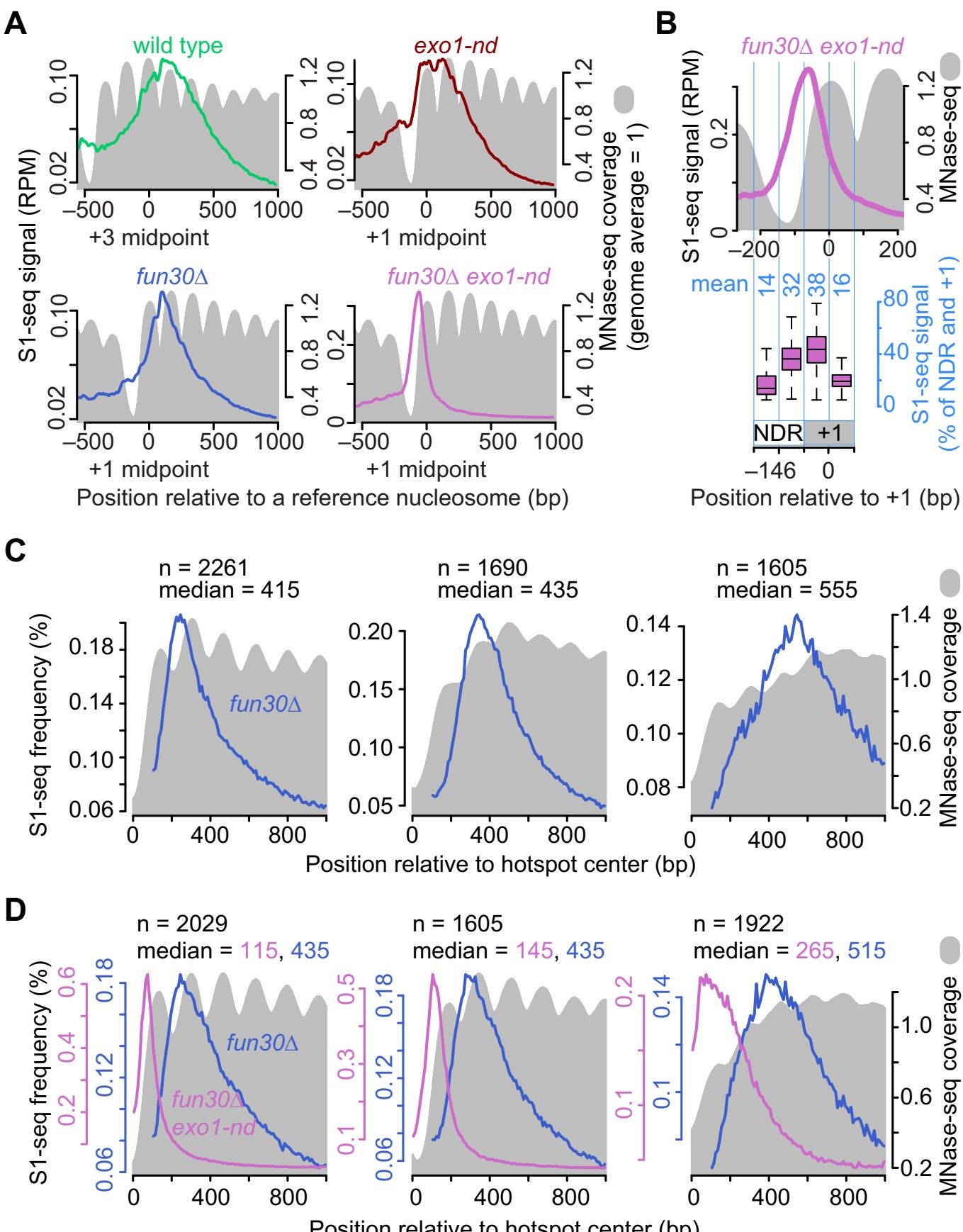

◄ **Figure 4.   S1-seq distribution relative to nucleosomes.**

(A) Average S1-seq resection signal (colored lines; 41-bp smoothed) and MNase-seq (gray filled; 0 h in meiosis in wild-type (Pan et al, 2011)) centered on midpoints of +3 or +1 nucleosomes with no other hotspots ≤3 kb downstream (n = 1815 for +3 nucleosomes and 1832 for +1 nucleosomes). The color coding for the genotype for the S1-seq profiles is preserved throughout this figure. (B) Tighter view of the *fun30Δ exo1-nd* resection profile from panel A. The boxplot below shows that substantial S1-seq signals overlap with the MNase signals within the +1 nucleosome. The total S1-seq reads in the +1 nucleosome (−73 to +73 bp relative to the midpoint) and NDR (-219 to -74) regions were taken as 100%, and the percentages of S1-seq in the four regions are shown in light blue numerals. Boxplot elements are as described in Fig. EV1C. (C, D) Resection tract lengths in the absence of Fun30 correlate with basal nucleosome occupancy. S1-seq profiles from *fun30Δ* (blue) or *fun30Δ exo1-nd* (magenta) are shown. Hotspots were grouped according to resection patterns using k-means clustering applied to spatial distributions of S1-seq signals in *fun30Δ* (C) or *fun30Δ exo1-nd* (D). The *fun30Δ* curves differ between (C, D) because the clusters contain different subgroups of hotspots in each panel. We used hotspots that were separated from their nearest neighboring hotspot by >1 kb and considered each side of each hotspot separately (2778 hotspots and 5556 hotspot sides). The S1-seq and premeiotic (t = 0 h) MNase-seq maps were averaged within each cluster and plotted as a function of distance from the hotspot midpoint. The number of hotspot sides in each cluster (n) and median resection lengths are indicated.

increases in ChIP-seq signal in response to DSBs (Fig. EV1B,C), which appears to be at least partly explained by the proximity of these features to Rec114 peaks (Fig. EV1D). Implications of DSB-stimulated Fun30 association with these sites away from hotspots are addressed in "Discussion".

## Fun30 remodels chromatin to facilitate efficient resection

In vitro, Fun30 slides nucleosomes and can exchange histone proteins in reconstituted nucleosomes (Awad et al, 2010; Byeon et al, 2013; Markert et al, 2021). In vivo, the *fun30Δ* mutation modestly alters nucleosome occupancies at centromeres and around promoters (Byeon et al, 2013; Durand-Dubief et al, 2012). Because Fun30 has chromatin remodeling activities, we sought evidence that its function in meiosis involves promoting the ability of the resection nucleases to act on nucleosomal substrates.

The NDRs of many yeast promoters are flanked by positioned nucleosomes, with the transcription start site (TSS) in the first (+1) nucleosome (Jansen and Verstrepen, 2011). To test if resection endpoints correlate spatially with preferred positions of nucleosomes, we averaged S1-seq signals around hotspots, using as a reference point the midpoints of +3 nucleosomes for wild type or +1 nucleosomes for the shorter resection tracts in the mutants (Fig. 4A). We then compared these population-average resection profiles to population-average chromatin structure as measured by sequencing of mononucleosomes released by digestion with micrococcal nuclease in wild-type cells (MNase-seq; Pan et al, 2011).

As previously shown (Mimitou et al, 2017), the averaged S1-seq signal in wild type was broadly distributed with a maximum between the +3 and +4 nucleosomes, while *exo1-nd* showed two peaks overlapping the +1 nucleosome and the linker between the +1 and +2 nucleosomes (Fig. 4A). Both profiles also showed modest scalloping in register with the edges of preferred nucleosome positions.

In contrast, the *fun30Δ* S1-seq showed a comparatively narrower distribution, with a sharp, scalloped peak between the +1 and +2 nucleosomes (Fig. 4A). The *fun30Δ exo1-nd* distribution was narrower still, forming a single smooth peak centered to the left of the MNase-seq peak for the +1 nucleosome and straddling the right edge of the NDR and the left edge of the +1 nucleosome (Fig. 4A,B). The S1-seq signal overlapped substantially with the +1 nucleosome position but tapered off sharply about halfway through (Figs. 4B and EV2). This pattern implies either that Mre11 can nick readily within the first half of the nucleosome, or that the +1

nucleosome is moved or destabilized in a Fun30-independent fashion after DSB formation. The results also show that nicking by Mre11 cannot spread readily beyond the first nucleosome in the absence of Fun30. Thus, in the absence of Fun30 there appears to be an even stronger correlation between chromatin structure and the distribution of resection endpoints.

To test this correlation further, we used k-means clustering to divide hotspots into groups that differed according to resection tract lengths in *fun30Δ* (Fig. 4C) or *fun30Δ exo1-nd* (Fig. 4D). We reasoned that if Fun30 normally mitigates nucleosomal barriers to resection, then these clusters should also have systematic differences in their average MNase-seq maps. Specifically, places where resection tends to go further even in the absence of Fun30 should be enriched for locations that intrinsically have a lower basal nucleosome occupancy, while places that tend to have shorter resection lengths should be those with higher nucleosome occupancy.

This analysis gave the predicted patterns. The cluster with the shortest resection tracts in *fun30Δ* had an average MNase-seq profile with pronounced peaks and high sequence coverage, indicating a tendency toward there being positioned nucleosomes of comparatively high occupancy (Fig. 4C, left). Conversely, the cluster with the longest resection tracts had markedly lower MNase-seq coverage that formed less well-defined peaks, indicating a tendency toward a more open chromatin structure with less regularly positioned nucleosomes and lower occupancy (Fig. 4C, right). Interestingly, the intermediate cluster had its highest MNase-seq coverage at 501 bp from the hotspot centers, just downstream of where most resection tracts in this cluster terminated (Fig. 4C, middle). These findings reinforce the correlation between Fun30-independent resection and constitutively open chromatin.

Similar results were observed for MRX/Sae2-mediated resection in the *fun30Δ exo1-nd* mutant. The cluster with the shortest resection tracts had well-defined peaks of high MNase-seq signal immediately flanking the hotspot NDR (Fig. 4D, left), whereas the cluster with longer resection tracts had more poorly defined and lower coverage MNase-seq peaks, consistent with more open and less regular chromatin structure (Fig. 4D, right). These observations further support that Fun30 helps the resection machinery overcome the inhibitory effects of nucleosomes.

This exercise also provided insight into the ability of Exo1 to resect nucleosomal DNA in the absence of Fun30. The narrow distribution of resection endpoints in clusters 1 and 2 in the *fun30Δ exo1-nd* mutant (Fig. 4D, left and middle) implies that nucleosomes are maintained in the absence of Fun30 and constrain the spread of

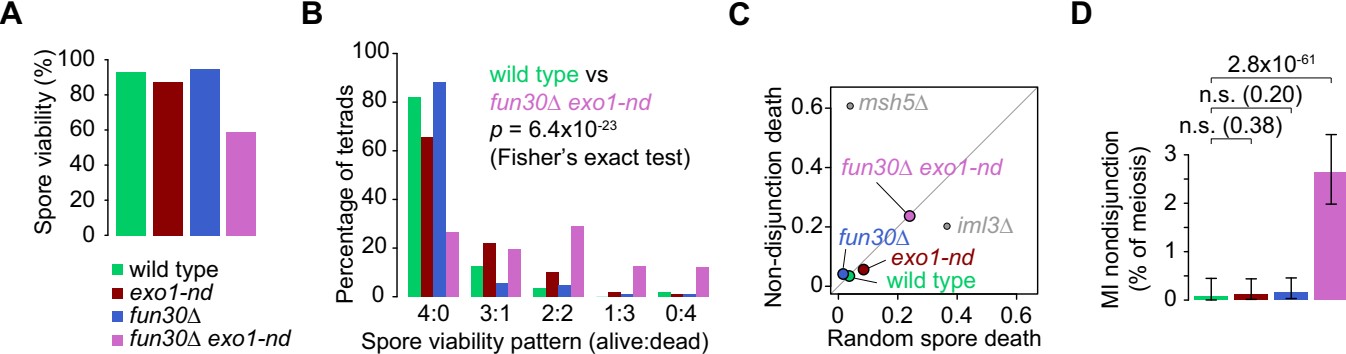

**Figure 5. Decreased spore viability and recombination defects in *fun30Δ exo1-nd*.**

(A) Spore viability of the indicated strains (*n* = 110 tetrads generated from mass mating followed by sporulation for each genotype (see methods), number of replicates = 1). (B) Spore viability patterns for the tetrads shown in (A). (C) Estimates of contributions from random spore death and MI nondisjunction to total spore death. Tetrad dissection data from (B) were subjected to the TetFit algorithm (Chu and Burgess, 2016; Chu et al, 2017). Data from *iml3Δ* and *msh5Δ* (Chu and Burgess, 2016) are shown for comparison. (D) MI nondisjunction (NDJ) frequency measured using spore-autonomous fluorescent markers (Thacker et al, 2011). The numbers of scored tetrads were 1241 (wild-type), 1644 (*exo1-nd*), 1913 (*fun30Δ*), and 2053 (*fun30Δ exo1-nd*). Error bars indicate 95% confidence intervals calculated by binomial distribution. Numbers above bar graphs indicate *P* values calculated by a two-sided exact binomial test (MI-NDJ). n.s. not significant (*P* > 0.05). Source data are available online for this figure.

nicking by MRX/Sae2. Nevertheless, when S1-seq profiles from *fun30Δ* were overlaid to show what the combined action of MRX/Sae2 and Exo1 could accomplish for these same hotspots, median resection lengths were ~300 nt longer than for MRX/Sae2 alone (blue traces in Fig. 4D). This implies that the same nucleosomes that constrain MRX/Sae2 can be overcome by Exo1.

## The *fun30Δ exo1-nd* mutant has reduced spore viability

The *fun30Δ exo1-nd* double mutant had decreased spore viability (58.8%), with an increased proportion of tetrads with two dead spores as well as smaller increases in tetrads with one, three, or four dead spores (Fig. 5A,B). By contrast, the *exo1-nd* single mutant had a more modest decrease in viability (87.3%) that was mostly attributable to an increase in three-spore viable tetrads, while the *fun30Δ* single mutant showed normal spore viability.

Tetrads with either two or four dead spores can arise from meiotic recombination defects that cause chromosome missegregation during the first meiotic division (MI nondisjunction: MI-NDJ) (Chu and Burgess, 2016). Tetrads with one or three dead spores are generally ascribed to random spore death (RSD), which can arise, for example, by defects in sister chromatid segregation. We used TetFit (Chu and Burgess, 2016; Chu et al, 2017) to estimate the relative contributions of RSD (24.0%) and MI-NDJ (23.7%) to spore death in the *fun30Δ exo1-nd* double mutant (Fig. 5C). This gives an estimated 4.0% MI-NDJ per chromosome. The increased frequency of MI-NDJ is consistent with a meiotic recombination defect, but the apparent mixture of RSD and MI-NDJ is in contrast to mutants with more unitary modes of spore death, such as *iml3Δ* (mostly RSD from defects in sister chromatid disjunction) or *msh5Δ* (mostly MI-NDJ from defects in homologous recombination) (Chu and Burgess, 2016).

We also measured the frequency of MI-NDJ of chromosome VIII using a spore-autonomous fluorescence assay (Fig. 5D) (Thacker et al, 2011). MI-NDJ was increased 32.6-fold by this assay to 2.63% in *fun30Δ exo1-nd*, consistent with the TetFit analysis.

## *fun30Δ exo1-nd* double mutants have compromised interhomolog recombination bias

To more rigorously evaluate recombination timing and efficiency, we used direct physical analysis of recombination intermediates and products at the strong *HIS4LEU2* hotspot. Genomic DNA was isolated from synchronized meiotic cultures, digested with appropriate restriction enzymes, separated by either one- or two-dimensional agarose gel electrophoresis, and analyzed by Southern blotting and indirect end labeling. Restriction site polymorphisms between the homologous chromosomes flanking the hotspot allow detection and quantification of DSBs; branched recombination intermediates (single-end invasions (SEIs) and double-Holliday junctions (dHJs)) between sister chromatids or between homologs; and both crossover and noncrossover recombination products (Fig. EV3A–D) (Hunter and Kleckner, 2001; Kim et al, 2010).

In the wild type, signals from DSBs, SEIs, and dHJs appeared and disappeared, crossover and noncrossover recombination products accumulated, and nuclear divisions occurred with the expected kinetics (Fig. 6A–G). More signal was observed for interhomolog dHJs (IH-dHJs) than for intersister (IS-dHJs) throughout meiotic prophase I (Fig. 6D,E, bottom), reflecting the normal bias in recombination partner choice that favors using homologous chromosomes rather than sister chromatids (Kim et al, 2010; Schwacha and Kleckner, 1997). The *exo1-nd* mutation increased the time-averaged amount of DSB signal (1.8-fold, estimated from comparative areas under the time-course curves) and recombination intermediates (1.7-fold for SEIs and 1.9-fold for total dHJs) but did not affect crossover levels and reduced the amount of noncrossovers slightly (Fig. 6A,B,D–G). Similar to previous findings (Zakharyevich et al, 2010), *exo1-nd* maintained interhomolog recombination bias, albeit somewhat weakened (Fig. 6D,E, bottom).

Unexpectedly, the *fun30Δ* mutation delayed the onset of DSB formation, the appearance of crossovers, and the completion of meiotic divisions, with each affected to a similar extent (~1 h) (Fig. 6A–C,E–G). These delays are likely attributable to slower

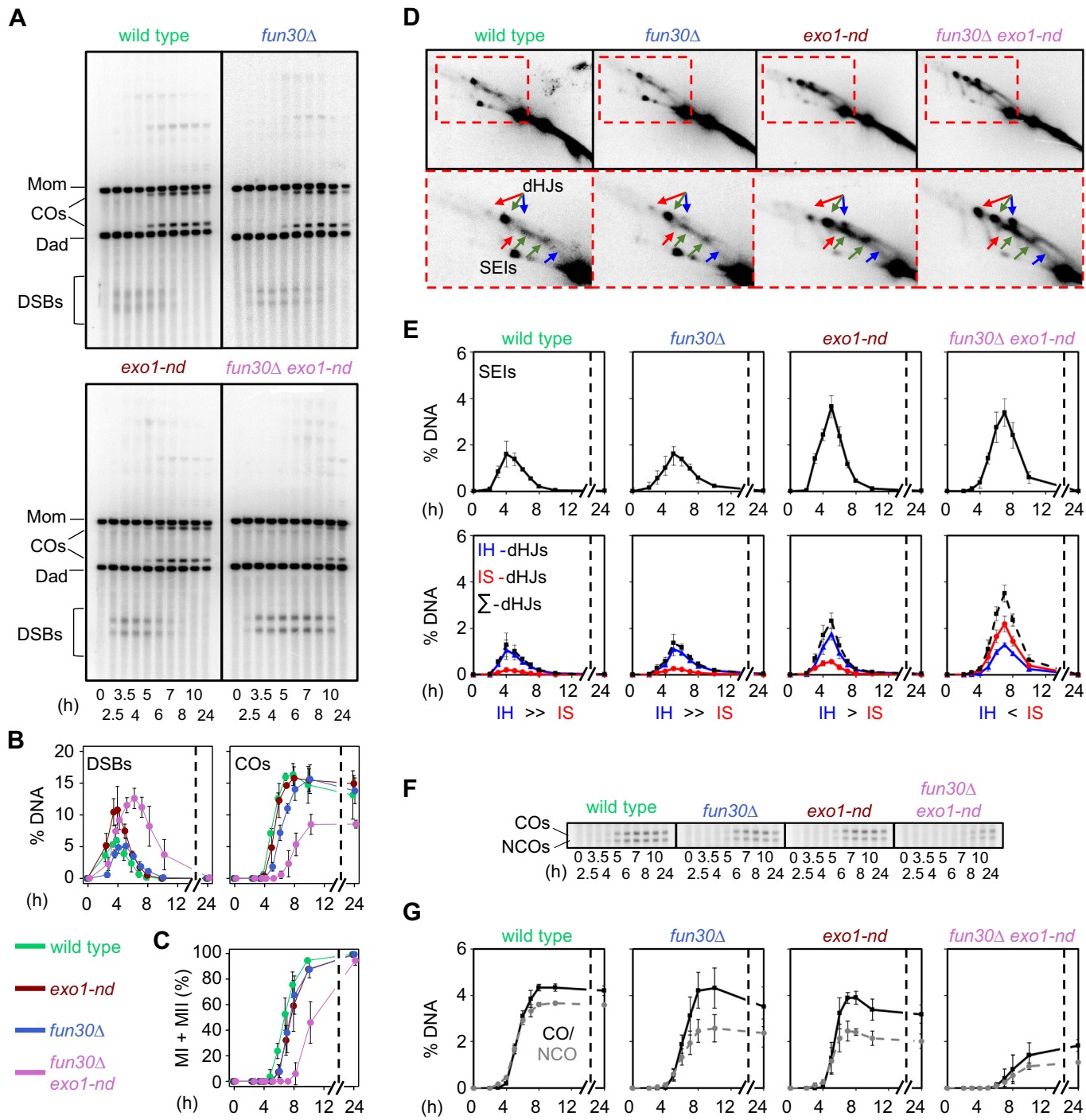

**Figure 6. Diminished interhomolog bias at *HIS4LEU2* in *fun30Δ exo1-nd* mutants.**

(A) One-dimensional (1D) gel analysis of DSBs and crossovers (COs) at the *HIS4LEU2* hotspot. (B) Quantification of DSBs and crossovers from 1D gel analysis. DSB and crossover levels are shown as a percentage of the total hybridization signal per lane. (C) Meiotic division time courses. The graph shows the percentage of cells that have completed one or both divisions. (D) Representative two-dimensional (2D) gels of SEIs and dHJs at *HIS4LEU2*. Arrows indicate interhomolog joint molecules (green) or Mom–Mom (red) and Dad–Dad (blue) intersister molecules (color coding as in Fig. EV3A). (E) Quantification of SEIs and dHJs from 2D gel analyses. In all graphs, the data are the mean ± SD for three independent meiotic cultures. (F) Representative gel images of crossovers and noncrossovers. (G) Quantification of crossovers and noncrossovers (mean ± SD for three independent meiotic cultures). Source data are available online for this figure.

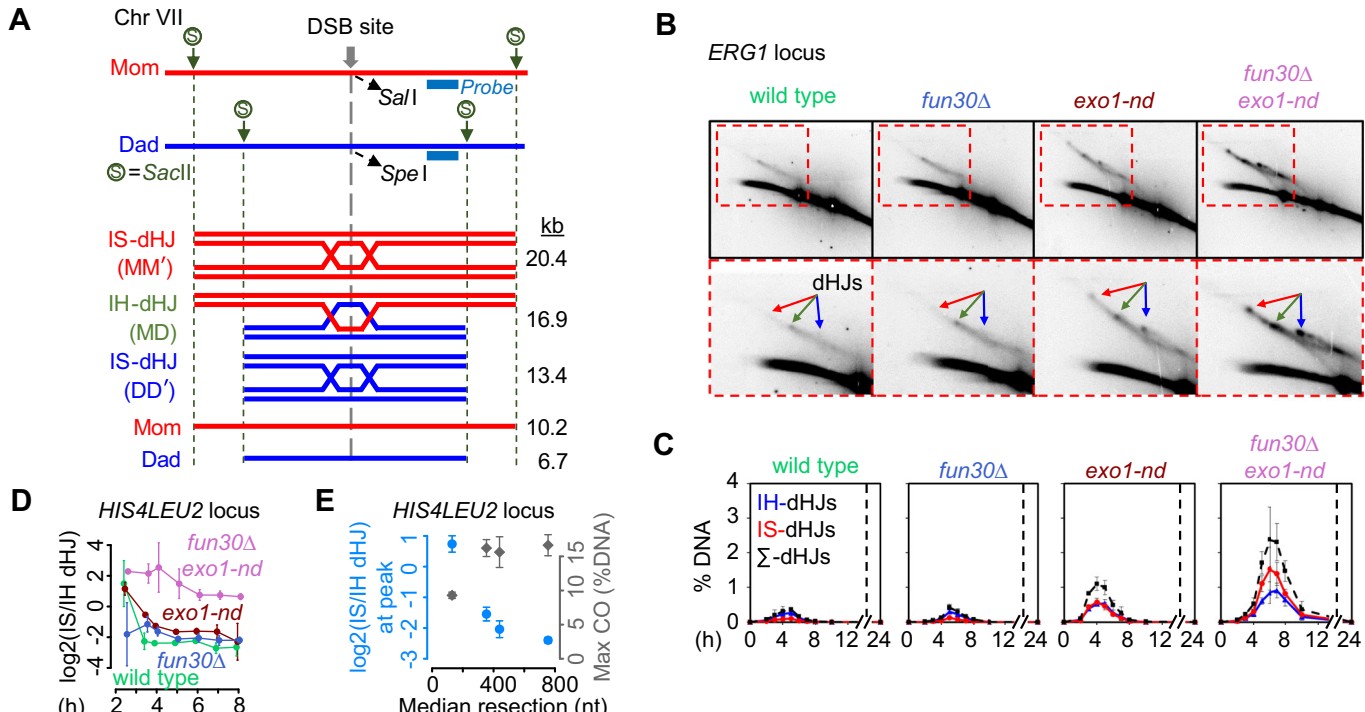

**Figure 7. Interhomolog bias at the *ERG1* hotspot.**

(**A**) Physical map of the *ERG1* hotspot on chromosome VII showing diagnostic *Sac*II restriction enzyme sites and the position of the Southern blot probe. Parental chromosomes were distinguished by *Sac*II restriction enzyme site polymorphisms. (**B**) Two-dimensional (2D) gel Southern blot analysis of SEIs and dHJs at *ERG1*. Arrows indicate interhomolog dHJs (green) and either Mom–Mom (red) or Dad–Dad (blue) intersister dHJs. (**C**) Quantification of dHJs at *ERG1* from 2D gel analysis. The data are the mean ± SD for three independent meiotic cell cultures. (**D**) Progression of interhomolog bias at *HIS4LEU2* throughout meiotic prophase. The strength of interhomolog bias was estimated by calculating the log$_2$-transformed ratio of IS-dHJ to IH-dHJ from the dataset in Fig. 6E (mean ± SD of three biological replicates). Smaller values indicate stronger interhomolog bias. (**E**) Different impacts of shortening resection tract length on interhomolog bias and crossover formation at the *HIS4LEU2* locus. Blue points show the log$_2$-transformed ratio of IS-dHJs to IH-dHJs calculated at the time point with maximal total dHJ signal for each individual replicate (wild-type: 4, 5, 4 h; *fun30*Δ: 5, 6, 5 h; *exo1-nd*: 4, 5, 5 h; *fun30*Δ *exo1-nd*: 7, 7, 7 h). Gray points show the average of the maximum crossover frequency for each time course. Values were calculated from the datasets in Fig. 6B,E. Error bars indicate mean ± SD. Source data are available online for this figure.

meiotic entry, as FACS analysis showed similarly delayed meiotic replication (~1 h) in *fun30*Δ compared to wild type (Appendix Fig. S2). IH-dHJs exceeded IS-dHJs in *fun30*Δ, indicating that strong IH bias was maintained, and final crossover levels were normal (Fig. 6B,E,G).

The *fun30*Δ *exo1-nd* double mutant combined several of the phenotypes of the single mutants. For example, meiotic DNA synthesis and DSB onset were delayed comparably to the *fun30*Δ mutant (Fig. 6A,B; Appendix Fig. S2), and peak levels of both DSBs and recombination intermediates were elevated similarly to *exo1-nd* (Fig. 6A,B,D,E). However, the double mutant also showed additional defects not seen with either single mutant, including prolonged presence of DSBs, which may indicate continued DSB formation and/or longer DSB lifespan (Fig. 6A,B); defective interhomolog bias (more IS-dHJs than IH-dHJs; Fig. 6D,E); substantially delayed (2.5–3 h) and reduced formation of both crossovers (decreased ~2-fold) and noncrossovers (decreased ~3-fold) (Fig. 6A,B,F,G); and greatly delayed (~3 h) meiotic divisions (Fig. 6C).

We verified the changes in DSB timing and/or levels for the single and double mutants at four natural hotspots (*ERG1*, *CYS3*, *BUD23*, and *ARG4*), observing similar results as at *HIS4LEU2*: DSBs were delayed in *fun30*Δ; elevated in *exo1-nd*; and delayed,

elevated, and persistent in *fun30*Δ *exo1-nd* (Fig. EV4). We also confirmed using two-dimensional gel analysis at the *ERG1* hotspot that the interhomolog bias of dHJs was maintained in *fun30*Δ, weakened in *exo1-nd*, and greatly reduced in *fun30*Δ *exo1-nd* double mutants (Fig. 7A–C).

We consider it likely that the pronounced defect in interhomolog bias in the *fun30*Δ *exo1-nd* double mutant is at least one cause of the persistent DSB signal, reduced number of crossovers, and strong delay in crossover and noncrossover formation. It is also likely that these recombination defects in turn cause delayed prophase I exit (resulting in greatly delayed meiotic divisions), increased MI-NDJ, and decreased spore viability (see "Discussion").

## Discussion

We previously proposed that one or more chromatin remodelers contribute to meiotic resection by removing nucleosomes in the vicinity of DSBs (Mimitou et al, 2017). Our findings here implicate Fun30 in this role in yeast. Chromatin remodeling contributes to resection in vegetative yeast and mammalian cells as well (Fowler et al, 2022; Peritore et al, 2021). However, unlike in vegetative cells, where Fun30 acts redundantly with other remodelers to promote

resection specifically over relatively long distances (Chen et al, 2012; Costelloe et al, 2012; Eapen et al, 2012), during meiosis Fun30 by itself strongly affects both initial (MRX/Sae2) and secondary (Exo1) resection steps. Fun30 thus plays a predominant role in DSB processing uniquely in the context of meiotic recombination.

We found no effects of the other chromatin remodeler mutations tested, either alone or in combination with *fun30Δ*. Because we used *swr1Δ* and *htz1Δ* null mutants, we can exclude significant contributions of Swr1 and Swr1-dependent deposition of Htz1. Fun30 thus may be the only relevant remodeler in meiosis, but it is important to note that we do not formally exclude contributions of other remodelers. Normal mitotic resection involves the nonessential INO80 subunit Arp8 (Tsukuda et al, 2005); although we found that Arp8 was dispensable for meiotic resection, we cannot rule out that INO80 has an Arp8-independent role in meiotic resection. Moreover, we were unable to test the involvement of RSC because deletion of its nonessential subunit Rsc2 or meiotic depletion of its essential subunit Sth1 (Peritore et al, 2021) interfered with meiotic entry in our strains, possibly because of RSC's role in *IME2* expression (Inai et al, 2007).

## Spatiotemporal coordination between DSB formation and resection

Meiotic chromosomes are organized into arrays of chromatin loops that are anchored at their bases by proteinaceous axial elements that include (among other things) cohesins and meiosis-specific axis proteins such as Red1 and Hop1 (Kleckner, 1996, 2006; Klein et al, 1999; Panizza et al, 2011; Smith and Roeder, 1997; Zickler and Kleckner, 1999). The DSB-forming machinery appears to localize to the axes but most DSBs occur in DNA segments that are usually in the loops. This paradox has led to a model in which tethered loop-axis complexes (TLACs) are formed by the recruitment of loop DNA to axes to allow DSB formation and subsequent recombination (Blat and Kleckner, 1999; Kleckner, 2006; Panizza et al, 2011) (Fig. EV5).

We therefore interpret the DSB-dependent enrichment of Fun30 at both hotspots and Rec114 peaks (which are thought to represent axis-associated assemblies of DSB-promoting factors) as reflecting the spatial coordination between hotspots and axes within the context of TLACs (Fig. EV5). We previously showed that histone H3 is phosphorylated on threonine 11 by the DSB-responsive kinase Mek1, and that phospho-H3 is enriched at both axis attachment sites and around hotspots (Kniewel et al, 2017). Fun30 thus provides another example of a protein whose chromatin localization in response to DSBs appears to be shaped by the higher-order organization of meiotic chromosomes.

Our ChIP-seq data indicate that meiotic DSBs provoke the recruitment of Fun30 nearby, likely on the broken chromatid and/or its sister chromatid. DSBs also activate a feedback control where local activation of the DSB-response kinase Tel1 (ortholog of mammalian ATM) inhibits Spo11 cleaving the same hotspot again (Johnson et al, 2021; Lange et al, 2011; Prieler et al, 2021). An interesting implication of post-DSB recruitment of Fun30 in cis is that the resulting chromatin remodeling occurs exclusively on the homolog that has lost DSB competence because of Tel1-mediated feedback control. By restricting chromatin changes to hotspots that are unlikely to experience more DSBs, Spo11 may thus be constrained to cut only where allowed by chromatin that has not been acted on by Fun30. This constraint would inhibit DSBs from forming within gene bodies, possibly affecting the mutagenic potential of recombination. Fun30 recruitment and activation at DSBs are cell cycle regulated in vegetative cells (Bantele and Pfander, 2019; Chen et al, 2016), suggesting that multiple regulatory strategies may have evolved to limit Fun30 remodeling in different contexts.

## Nucleosomes as a resection barrier in vivo

In vitro, resection by Exo1 is almost completely blocked by a nucleosome (Adkins et al, 2013), and MRX/Sae2 incision can occur directly adjacent to but not apparently within a nucleosome (Wang et al, 2017). Genomic studies also suggest inhibition of MRX/Sae2 incision by nucleosomes in vegetative cells (Gnugge et al, 2023). We found here that constitutive chromatin structure strongly affected the residual Fun30-independent resection that could be carried out by MRX/Sae2 alone or in combination with Exo1, supporting the interpretation that nucleosomes pose barriers to both resection nucleases during meiosis as well (Mimitou et al, 2017). Interestingly, however, our findings also indicate that nucleosomes are unlikely to be an absolute block to the nucleases.

First, we found that MRX/Sae2 by itself (*fun30Δ exo1-nd*) frequently cleaves within the position usually occupied by the +1 nucleosome rather than being strictly constrained to the NDR and the linker between nucleosomes. The DNA cleavage positions in this case mostly fall within the first half of the nucleosome position and rarely spread up to the next nucleosome, so we can rule out that these cleavage events are solely due to the +1 nucleosome having been removed entirely in all or a subset of cells.

Second, we found that Exo1 still carries out substantial resection in the absence of Fun30 (comparing *fun30Δ* to *fun30Δ exo1-nd*). This means that Exo1 is able to digest DNA within the same nucleosomes that are constraining MRX/Sae2 and causing resection tracts to shorten in *fun30Δ exo1-nd* compared to *exo1-nd* alone.

We therefore conclude that both MRX/Sae2 and Exo1 have some limited ability to digest nucleosomal DNA in vivo without intervention by Fun30, even though they appear incapable of doing so in vitro (Adkins et al, 2013; Wang et al, 2017). One possibility is that both enzymes can act on nucleosomal DNA directly. We note that biochemical studies to date have relied on the strong artificial 601 nucleosome positioning sequence, which provides an exceptionally stable nucleosome. Perhaps both nucleases are better able to act on nucleosomes of more physiological stability. Alternatively, there may be Fun30-independent processes that destabilize nucleosomes. We can rule out Swr1 or Htz1, but remaining nonexclusive candidates for such processes could include transcription, the action of remodelers other than Fun30 or Swr1, histone posttranslational modifications, and/or DSB-provoked changes in chromatin structure (e.g., caused by topological changes or DSB-response signaling).

Our findings indirectly support that Fun30 remodels chromatin near DSBs to promote resection, but we currently lack the ability to directly measure nucleosome position and occupancy specifically on broken chromatids, leaving some molecular details unclear. For example, it is uncertain if nucleosomes near DSBs are altered by Fun30-independent factors (e.g., DSB formation per se or MRX/Sae2 nicking). Moreover, the specific types of nucleosome

remodeling caused by Fun30 (sliding, eviction, changes in histone–DNA binding strength) also remain to be determined. At even the strongest natural hotspots, only a small minority of the chromatids in a population of cells is broken, so it is challenging to assay chromatin structure specifically on broken chromatids using conventional bulk-population methods. Developing novel methods to exclusively determine chromatin structure on broken chromatids would shed light on these issues.

## Minimum resection length required for meiotic recombination

In vegetative cells, recombination can occur with homology lengths less than 40 bp, and normal recombination efficiency is estimated to need only ~100–250 bp on each side of the DSB (Hua et al, 1997; Inbar et al, 2000; Ira and Haber, 2002; Jinks-Robertson et al, 1993; Manivasakam et al, 1995). It has been suggested that the reason meiotic resection in wild-type is considerably longer than these minima is to ensure that all DSBs are resected enough to allow recombination to proceed efficiently and accurately (Zakharyevich et al, 2010). It was further noted that most DSBs are resected further than these minima in an *exo1-nd* mutant, and only a small fraction of DSBs had resection tracts short enough to potentially impair recombination (Zakharyevich et al, 2010).

Our findings refine these ideas and suggest important roles of resection aside from simply providing enough ssDNA for homology searching. First, we note that the *fun30Δ* single mutation reduced median resection lengths almost as much as *exo1-nd* alone, but gave better spore viability. One important difference between these single mutants is that *fun30Δ* cells had a longer minimum resection distance than *exo1-nd*, such that fewer resection tracts were very short (e.g., only ~4.6% were <200 nt long in *fun30Δ*, vs. 8.4% in *exo1-nd*). We surmise that the relative paucity of extremely short resection tracts is the reason for better preservation of successful meiosis in *fun30Δ* mutants than in *exo1-nd*, measured as better spore viability.

Second, the *fun30Δ exo1-nd* double mutant had extremely shortened resection as well as recombination defects leading to increased rates of homolog missegregation and spore death. We do not exclude the possibility that there are resection-independent defects contributing to these phenotypes, e.g., altered gene expression (Byeon et al, 2013; Durand-Dubief et al, 2012). (Altered gene regulation is a good candidate to explain the delayed meiotic entry in the *fun30Δ* background.) However, it is plausible that the meiotic recombination defects in the double mutant are mostly (or even entirely) a consequence of the shortened resection itself. If so, and if the principal importance of resection length is to provide enough ssDNA for efficient homology searching, we initially expected to observe fewer and/or less stable recombination intermediates. Unexpectedly, though, both SEIs and dHJs were abundant, suggesting that strand exchange and its prerequisite homology search were both still effective despite the greatly reduced ssDNA length. Instead, we observed a pronounced decrease in interhomolog bias.

Decreased bias probably explains, at least partially, the reduced recombination. However, the *fun30Δ exo1-nd* double mutant had a decrease in crossovers that was more severe than the decrease in interhomolog joint molecules (SEIs and IH-dHJs). Because these recombination intermediates are crossover-committed in normal

meiosis (Hunter and Kleckner, 2001; Kim et al, 2010; Schwacha and Kleckner, 1997), it may be that the double mutant also has a further defect in crossover maturation. We also infer that the overall reduction in recombination is the cause of the increased chromosome missegregation.

Third, as reported previously (Joshi et al, 2015), wild type progressively establishes interhomolog bias during meiotic prophase (Fig. 7D). In contrast, *exo1-nd* showed delays in the establishment of interhomolog bias in addition to reducing overall bias, and *fun30Δ exo1-nd* failed almost entirely to establish the bias. The median resection length negatively correlated with the degrees of interhomolog bias (blue points in Fig. 7E). Interestingly, however, crossover formation showed a more pronounced threshold effect, with significant defects only apparent with the most extreme reduction in resection length (gray points in Fig. 7E). We suggest that homeostatic mechanisms that control crossover outcomes (e.g., crossover homeostasis (Martini et al, 2006)) provide robustness in the face of relatively modest defects in interhomolog bias. There appears to be a threshold between 17 and 47% of wild-type ssDNA content, below which overt crossover defects begin to materialize.

We envision two reasons for the observed decrease in interhomolog bias when resection is very short. One possibility is that there is a minimal requirement for ssDNA-provoked DNA damage signaling. The establishment of interhomolog bias involves the activation of Mec1 and Tel1 kinases in response to DSB formation, which in turn activates Mek1 kinase (Hollingsworth and Gaglione, 2019). Because Mec1 activation requires ssDNA-bound RPA (Zou and Elledge, 2003), reducing the total ssDNA content by shortening resection without decreasing DSB numbers could reduce interhomolog bias similar to the effect of having fewer DSBs (Joshi et al, 2015). A second, nonexclusive possibility is that very short resection tracts might compromise the necessary loading of both Dmc1 and Rad51. Interhomolog bias is defective in the absence of Rad51, which plays a strand-exchange-independent role in promoting normal partner choice (Cloud et al, 2012; Hong et al, 2013; Lao et al, 2013; Schwacha and Kleckner, 1997). Perhaps extremely short resection tracts sometimes fail to load sufficient Rad51, thereby compromising interhomolog bias. Regardless of which of these scenarios is correct, our results strongly indicate that a major constraint shaping how long meiotic resection needs to be is the minimal amount of ssDNA necessary to achieve appropriate regulation of recombination partner choice.

## Methods

**Reagents and tools table**

| Reagent/resource | Reference or source | Identifier or catalog number |
|---|---|---|
| **Experimental models** | | |
| SK1 (*S. cerevisiae*) | Kane and Roth, 1974 | N/A |
| IFO1815 (*S. mikatae*) | Scannell et al, 2011 | N/A |
| **Recombinant DNA** | | |
| pFA6a-13Myc-KanMX6 | Longtine et al, 1998 | N/A |
| **Antibodies** | | |
| Anti-Myc (9E10) | BioXCell | BE0238 |

| Reagent/resource | Reference or source | Identifier or catalog number |
|---|---|---|
| **Oligonucleotides and other sequence-based reagents** | | |
| PCR primers | This study | "Methods" |
| **Chemicals, enzymes, and other reagents** | | |
| antifoam 204 | SIGMA | A6426 |
| zymolyase 20 T | US Biological | Z1000 |
| zymolyase 100 T | US Biological | Z1004 |
| trioxsalen | SIGMA | T1637 |
| β-agarase I | NEB | M0392S |
| CHEF Disposable Plug Molds | BioRad | 1703713 |
| Protease inhibitor cocktail | SIGMA | P8215 |
| zirconia/silica beads (0.5 mm) | Biospec | 11079105z |
| UltraKem LE agarose | Young Science | Y50004 |
| Seakem Gold agarose | Lonza | 50150 |
| Biodyne B membrane | Pall | 60201 |
| **Software** | | |
| Quantity One | BioRad | |
| R (RStudio version 1.0.143, R version 4.0.3) | https://www.r-project.org/ | |
| **Other** | | |
| E220 evolution Focused-ultrasonicator | Covaris | 500429 |
| FastPrep-24 | MP Biomedicals | 116004500 |
| HiSeq2000 | Illumina | |

## Yeast strain and plasmid construction

Unless otherwise noted, all yeast strains used in this study (Appendix Table S1) were of the SK1 background (Kane and Roth, 1974). We used a standard lithium acetate method (Gietz and Schiestl, 2007) and verified transformants by PCR and Southern blotting. The *fun30Δ*, *arp8Δ*, *htz1Δ*, and *swr1Δ* mutants were generated by replacing the coding sequences with a G418 resistance cassette (*KanMX4*). Appropriate crosses and tetrad dissection were then used to generate single and double mutants. The *exo1-nd* (D173A) mutation creates a *Drd*I restriction enzyme site that was used to follow the *exo1-nd* allele in crosses. Spore-autonomous fluorescent markers (*THR1::m-Cerulean-TRP1* and *CEN8::tdTomato-LEU2*, *ARG4::GFP\*-URA3*) (Thacker et al, 2011) were introduced to the *fun30Δ exo1-nd* mutant by crossing and tetrad dissection.

For ChIP-seq, Fun30 was C-terminally tagged by integrating a DNA fragment containing 13 copies of the Myc epitope and the *KanMX6* cassette amplified from pFA6a-13Myc-KanMX6 (Longtine et al, 1998) before the stop codon of the *FUN30* open reading frame. The primer sets (uppercase: 50 bp homology sequences, lowercase: annealing sequences) are as follows: 5′-TGGAGGATA-TAATTTATGATGAAAACTCGAAACCGAAGGGAACCAAA-GAAggtggtggtggtggtggtggtggtCGGATCCCCGGGTTAATTAA; 5′-

TTTATTTTCTGCTTATCTATTTACTTTTTTACTATATTTT-TATTTATTTActggatggcggcgttagtatcgaatcgacagcagtatagcgacc.

## Sporulation

### Meiotic cultures for S1-Southern and S1-seq

Yeast strains were sporulated as previously described (Mimitou and Keeney, 2018). We performed presporulation culture in YPA (1% yeast extract, 2% Bacto peptone, 2% potassium acetate, 0.001% antifoam 204 (Sigma)) for 14 h, then transferred to sporulation medium (2% potassium acetate with amino acids and 0.001% polypropylene glycol). Meiotic cells from 66 ml sporulation medium harvested at 4 h after transfer to sporulation medium were washed with 50 ml of water and 50 mM EDTA pH 8.0 and stored at −80 °C.

### Tetrad dissection

To avoid the accumulation of lethal mutations, we avoided prolonged culture of diploid strains where possible. Instead, haploid strains were freshly mated on YPD plates for 6 h and immediately sporulated in sporulation medium for >48 h. Tetrads were treated with 100 µg/ml zymolyase 20 T (US Biological) at 30 °C for 20 min and dissected on YPD plates. Spore viability was scored after 2 days of incubation at 30 °C.

### Meiotic time courses for physical analysis of recombination

Procedures were as described previously (Hong et al, 2019; Hong et al, 2013; Kim et al, 2010; Yoon et al, 2016). Yeast diploid cells were patched onto a YPG plate (1% yeast extract, 2% peptone, 2% agar, and 3% glycerol) and incubated at 30 °C for 12 h. Cells were then streaked on a YPD plate (1% yeast extract, 2% peptone, 2% agar, and 2% glucose) and incubated at 30 °C for two days. Single colonies were picked and inoculated in YPD liquid medium (1% yeast extract, 2% peptone, and 2% glucose) and cultured at 30 °C for 24 h. To synchronize yeast cell cultures, 400 µl of the YPD-cultured cells were diluted in 200 ml of pre-warmed SPS medium (0.5% yeast extract, 1% peptone, 0.17% yeast nitrogen base without amino acids, 0.5% ammonium sulfate, 1% potassium acetate and 50 mM potassium biphthalate; pH was adjusted to 5.5 with 10 N KOH) and grown at 30 °C for 17–18 h. Meiosis was induced by culturing cells in 250 ml sporulation medium (SPM; 1% potassium acetate, 0.02% raffinose, and 2 drops of antifoam per liter) at 30 °C. SPM-cultured cells were harvested at 0, 2.5, 3.5, 4, 5, 6, 7, 8, 10, and 24 h and then cross-linked with 0.1 mg/mL trioxsalen (Sigma, T1637) under 365-nm ultraviolet light for 15 min.

## S1-Southern and S1-seq

Procedures for the agarose plug preparation, S1 nuclease treatment and the subsequent Southern blotting or S1-seq library preparation were described previously (Mimitou and Keeney, 2018; Mimitou et al, 2017). These procedures were followed here with the following modifications; section numbers refer to (Mimitou and Keeney, 2018): Section 4.4.2, steps 3–5: Each cell pellet was resuspended in 700 µl 50 mM EDTA pH 8.0. Seven hundred microliter of the cell suspension was mixed with 238 µl Solution 1 (SCE: 1 M sorbitol, 0.1 M sodium citrate, 60 mM EDTA pH 7.0

containing 5% β-mercaptoethanol and 1 mg/ml zymolyase 100 T) at 40 °C and aliquoted to plug molds (BioRad) to prepare 20 agarose plugs. Section 5.2.2, steps 1–3: GELase was replaced by β-agarase I (NEB). The reaction (1 U per plug) was carried out at 42 °C. Section 5.2.2, step 23: The sonication step was carried out using Covaris E220 evolution with the settings PIP = 175 w, DF = 10%, CPB = 200, time: 180 s at 4 °C. Less than 130 µl sample per microtube was loaded in microTUBE AFA Fiber Crimp-Cap on Rack E220e 8 microTUBE Strip. Section 5.4.2, step 6: For the post-PCR step, instead of ethanol precipitation with ammonium acetate, we purified library DNA using QIAquick PCR Purification Kit (Qiagen) with a final elution volume of 30 µl warm 10 mM Tris-HCl pH 8.0. Section 5.4.2, step 11: For the size selection step, instead of 5× TBE PROTEAN gel, samples were run on 1.5% agarose at 100 volts for 2 h. The gel region from 200 to 700 bp was excised and DNA was purified by QIAquick Gel Extraction Kit (Qiagen) following the manufacturer's instructions (we included step 6 in the Quick-Start Protocol for QIAquick: add 500 µl Buffer QG to the QIAquick column and centrifuge for 1 min). Library DNA was eluted in 25 µl warm 10 mM Tris-HCl pH 8.0.

### Chromatin immunoprecipitation for Fun30-Myc

ChIP experiments were performed as described previously (Murakami and Keeney, 2014), with modifications in shearing chromatin and calibrating datasets. Similar to the previously described calibrated ChIP method (Hu et al, 2015), we used *S. mikatae* cells as a spike-in control to compare ChIP-seq signals between datasets. For each *S. cerevisiae* strain, cells were sporulated using the YPA presporulation protocol as described above. Fifty milliliters ($2 \times 10^9$ cells) of culture at 4 h in meiosis was fixed with 1% formaldehyde for 15 min at room temperature, with mixing at 50 rpm. Cross-linking was quenched by adding glycine to 131 mM for 5 min. Cells were washed twice with 20 ml cold TBS buffer, frozen with liquid nitrogen, and stored at −80 °C until further steps. *S. mikatae* cells were sporulated using the SPS presporulation protocol (Murakami et al, 2020). Cells harvested at 4 h in meiosis were fixed and washed with the same condition described above. An aliquot of $2 \times 10^8$ *S. mikatae* cells (10% of the number of *S. cerevisiae* cells) was added to each sample.

Cells were resuspended in FA lysis buffer (50 mM HEPES-NaOH pH 7.5, 150 mM NaCl, 2 mM EDTA, 1% Triton X-100, 0.1% sodium deoxycholate, 7 µg/ml aprotinin, 10 mg/ml each of leupeptin, pepstatin A, and chymostatin, 1 mM PMSF, and 1 × each of Roche phosphatase and 1% protease inhibitor cocktails [Sigma]) (de Jonge et al, 2020; Vale-Silva et al, 2019) in 2-ml screw-cap Eppendorf tubes, and disrupted using zirconia/silica beads (0.5 mm, Biospec Products; ~900 µl per sample) and a FastPrep-24 (MP Biomedicals) with ten rounds of vigorous shaking at 6.5 m/s for 60 s. Lysates were pelleted by centrifugation at 15,000 rpm for 5 min at 4 °C. Chromatin in the whole-cell extracts was sheared by sonication using Covaris E220 evolution. To yield an average DNA size of around 350 bp (range 100–500 bp), 1 ml whole-cell extract with SDS added to 0.1% final concentration (Pchelintsev et al, 2016) was loaded in miliTUBE with AFA fiber and assembled on miliTUBE holder. Sonication conditions were 140 w, DF = 5%, CPB = 200, time = 8 min at 4 °C. Sonicated chromatin was centrifuged at 15,000 rpm for 5 min at 4 °C, and the supernatant (input) was collected for immunoprecipitation steps using anti-myc

antibody (9E10, RRID:AB_2857941) as described (Murakami and Keeney, 2014). DNA purified from input and immunoprecipitate samples was further sonicated prior to library preparation at the genomics core facility (Integrated Genomics Operation, Memorial Sloan Kettering Cancer Center).

### Physical analysis of recombination intermediates

Sporulation conditions and subsequent molecular biology procedures to detect recombination intermediates were as described previously (Kim et al, 2010; Oh et al, 2009). Genomic DNA preparation has been described (Hong et al, 2019; Hong et al, 2013; Kim et al, 2010; Yoon et al, 2016). Cells were treated with Zymolyase (US Biological) and then lysed in guanidine-HCl solution (4.5 M guanidine-HCl, 0.1 M EDTA, 0.15 M NaCl, and 0.05% sodium lauroyl sarcosinate) at 65 °C for 15 min. Genomic DNA was extracted twice with phenol/chloroform/isoamyl alcohol (25:24:1) and precipitated with ethanol. The DNA pellets were washed with 70% ethanol and dried at 4 °C overnight. Meiotic recombination was analyzed by gel electrophoresis at *HIS4LEU2* and *ERG1* loci similarly. Physical DNA analysis at the *HIS4LUE2* locus on chromosome III was performed as described previously (Hong et al, 2019; Hong et al, 2013; Kim et al, 2010; Lee et al, 2021; Yoon et al, 2016). Parental homologs are distinguished via *Xho*I restriction site polymorphism. Genomic DNA (2 µg) was digested with *Xho*I or *Xho*I plus *Ngo*MIV for one-dimensional (1D) gel analysis. The *ERG1* locus on chromosome VII was detected to obtain the DNA species of DSBs, and joint molecules by *Sac*II restriction site polymorphisms (Lao et al, 2013; Lee et al, 2021; Thacker et al, 2014). The DNA samples were loaded onto 1D gels (0.6% UltraKem LE agarose (Young Science) in 1 × Tris-borate-EDTA buffer), and electrophoresis was carried out in a 1 × Tris-borate-EDTA buffer for 24 h. For two-dimensional (2D) gel analysis, genomic DNA (2.5 µg) was digested with *Xho*I for the *HIS4LEU2* locus and *Sac*II for the *ERG1* locus. The DNA samples were loaded onto 1D gels (0.4% Seakem Gold agarose (Lonza) in 1 × Tris-borate-EDTA buffer), and electrophoresis was carried out in 1 × Tris-borate-EDTA buffer for 21 h. The 1D gel was stained with 0.5 µg/ml ethidium bromide (EtBr), and the gel strips of interest were cut and placed in a 2D gel tray. The gel electrophoresis was carried out in 1 × Tris-borate-EDTA buffer in 4 °C cold room. The gels were transferred to Biodyne B membrane (Pall). For Southern blot analysis, hybridization was carried out using probes labeled with [α-³²P]-dCTP (Lao et al, 2013; Lee et al, 2021; Thacker et al, 2014). Hybridization signals were visualized using a phosphoimager and quantified using Quantity One software (BioRad).

### Spore-autonomous fluorescent reporter assay

Yeast cells were sporulated for two days and subjected to subsequent analysis to measure genetic distance and MI nondisjunction frequencies on chromosome VIII as described previously (Thacker et al, 2011) and in Appendix Tables S2 and S3. Surprisingly, crossing over was not reduced in the *fun30Δ exo1-nd* double mutant compared to the *exo1-nd* single mutant for the intervals on chromosome VIII (Appendix Table S2). The two intervals examined (*CEN8–ARG4* and *ARG4–THR1*) differed in their response to the mutations: while the *exo1-nd* and *fun30Δ exo1-nd* mutants both

showed decreased genetic distance in the *CEN8–ARG4* interval, the *fun30Δ* mutant instead showed increased genetic distance specifically in the *ARG4–THR1* interval. The reason for the differences between intervals and between these data and the physical analysis at *HIS4LEU2* is not known, but the results may suggest a chromosomal position effect for recombination alterations tied to changes in resection. Other examples of inconsistencies between intervals for effects of mutations on genetic distance have been observed in other studies (Zakharyevich et al, 2010).

## Bioinformatic analysis

### S1-seq mapping and analysis

Sequencing (50-bp paired-end reads; Illumina HiSeq2000) was performed in the MSK Integrated Genomic Operation. In silico clipping of library adaptors and mapping to the genome was performed by the MSK Bioinformatics Core Facility using a custom pipeline as described (Mimitou and Keeney, 2018) with modifications. The code for read processing and mapping is available online at https://github.com/soccin/S1Seq. After mapping, the reads were separated into unique and multiple-mapping sets, but only uniquely mapping reads were analyzed in this study.

All downstream analyses were performed using R (RStudio version 1.0.143, R version 4.0.3 GUI 1.73 Catalina build). Map curation before analysis was performed by masking DNA ends and meiotic DSB-independent reads, with mitochondrial DNA and the 2-micron plasmid excluded as described (Mimitou et al, 2017). A global view of meiotic double-strand break-end resection and the mask coordinates can be found at https://github.com/soccin/S1Seq. Each map was normalized to reads per million remaining mapped reads and the biological replicates were averaged.

We used a hotspot list compiled from a combination of multiple independent wild-type Spo11-oligo maps (Mohibullah and Keeney, 2017). Different from (Mimitou et al, 2017), the left arm of chromosome 3 was not censored because all strains used for S1-seq are without *HIS4LEU2* and *leu2::hisG* artificial hotspots on this chromosome arm. Published maps were used of nucleosome occupancy (Pan et al, 2011), nucleosome midpoints (Zhang et al, 2011), and S1-seq in a *spo11* mutant with altered DSB locations (Claeys Bouuaert et al, 2021).

### Calibrated Fun30 ChIP-seq

Paired-end 50-bp reads were filtered and end trimmed, followed by removal of the reads containing tag sequences, and then mapped to a combined "genome" consisting of the *S. cerevisiae* (sacCer2, strain S288c-derived from SGD (*Saccharomyces* Genome Database)) and *S. mikatae* (IFO1815 (Scannell et al, 2011)) reference genomes. The sequence identities between *S. cerevisiae* and *S. mikatae* are 84% and 70% within the genic and intergenic regions, respectively (Kellis et al, 2003). We use sacCer2 for mapping the *S. cerevisiae* reads even though the samples are from the SK1 strain background to maintain consistency with prior data, and because sequence polymorphisms between the biological sample and the reference genome have little effect on mapping ability (Pan et al, 2011).

Each *S. cerevisiae* coverage map was normalized according to the *S. mikatae* read density for the same antigen from the same culture. Coverage maps generated from *S. cerevisiae* reads as previously described (Murakami and Keeney, 2014) were divided by the total

number of reads that were uniquely mapped to *S. mikatae* chromosomes to create input and immunoprecipitate maps normalized to spike-in control. Each normalized immunoprecipitate map was divided by the corresponding normalized input map to generate a ChIP coverage map normalized to the spike-in control. The *S. mikatae* spike-in control minimizes the effects of sample-to-sample variation during lysis, immunoprecipitation, and library preparation. Also, the fixed ratio of *S. cerevisiae* to *S. mikatae* is an excellent scaling factor for comparing the amount of Fun30 enrichment between samples for the same antigen (Hu et al, 2015). We used ARS and tRNA coordinates (sacCer2) downloaded from Saccharomyces Genome Database (yeastgenome.org).

### K-means clustering (Fig. 4C,D)

To investigate the relationship between pre-existing chromatin structure (i.e., the chromatin structure before DSB formation or DSB-provoked remodeling at hotspots) and the resection patterns in *fun30Δ* mutants, we used k-means clustering to group hotspots by their resection endpoint distributions. To do so, we selected hotspots with no other hotspots located within 1 kb ($n = 2778$) to avoid confounding effects of resection from neighboring DSBs. For each hotspot, we took S1-seq signals within 1 kb to the right (top strand reads) and left (bottom strand reads) of the hotspot midpoint, resulting in a total of 5556 resection profiles. We subtracted the background from each profile (defined as the lowest S1-seq signal within each profile), then normalized each profile to its total signal. This normalization was done to remove differences in DSB frequency and focus the clustering exercise on the spatial distribution rather than signal strength. The collection of S1-seq profiles from each dataset (*fun30Δ* single mutant or *fun30Δ exo1-nd* double mutant) was then clustered into three groups using the kmeanspp function in the LICORS package in R (https://CRAN.R-project.org/package=LICORS). For each group, the average of the resection profiles was overlaid with the averaged nucleosome data for the corresponding genomic locations (Pan et al, 2011).

### TetFit

We estimated the frequencies of MI nondisjunction death and random spore death based on finding the best-fit distribution of tetrads with 4, 3, 2, 1 and 0 viable spores to an observed distribution using the R algorithm TetFit (Chu and Burgess, 2016) with the default parameters ndint = 500, rsdint = 500, chr = 16, anid = 0.035, ndm = 10, minrsd = 0.0, maxrsd = 0.8, minnd = 0.0, maxnd = 0.017.

## Data availability

The datasets produced (Appendix Table S4) and analyzed in this study are available in the following databases: Fun30 ChIP-seq data: Gene Expression Omnibus GSE221033; S1-seq data: Gene Expression Omnibus GSE221377; Rec114 ChIP-seq data (Murakami and Keeney, 2014): Gene Expression Omnibus GSE52970; MNase-seq data (Pan et al, 2011): Gene Expression Omnibus GSE26452; S1-seq data (Claeys Bouuaert et al, 2021): Gene Expression Omnibus GSE150313.

The source data of this paper are collected in the following database record: biostudies:S-SCDT-10_1038-S44318-024-00318-8.

# Peer review information

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

## Acknowledgements

This article is subject to the Open Access to Publications policy of the Howard Hughes Medical Institute (HHMI). HHMI lab heads have previously granted a nonexclusive CC BY 4.0 license to the public and a sublicensable license to HHMI in their research articles. Pursuant to those licenses, the author-accepted manuscript of this article can be made freely available under a CC BY 4.0 license immediately upon publication. The authors thank A Viale and N Mohibullah of the Memorial Sloan Kettering Cancer Center (MSK) Integrated Genomics Operation (IGO) for DNA sequencing and N Socci at the MSK Bioinformatics Core Facility for mapping ChIP-seq and S1-seq reads. MSK core facilities are supported by NCI Cancer Center Support Grant P30 CA008748. The IGO was further funded by the Cycle for Survival and the Marie-Josée and Henry R. Kravis Center for Molecular Oncology. The authors thank members of the Keeney laboratory, especially S Yamada for advice on data analysis and S Kim for sharing unpublished information. The authors thank M Lichten for discussion, M Neale and V Garcia for sharing unpublished information, and N Hunter for strains and plasmids. This work was supported by NIH grant R35 GM118092 to SK, National Research Foundation of Korea grant funded by the Korea government (MSIT) 2022M3A9J4079468 to KPK, and Medical Research Council Career Development Award MR/W027313/1 to HM.

## Author contributions

**Pei-Ching Huang**: Conceptualization; Formal analysis; Investigation; Visualization; Writing—original draft; Writing—review and editing. **Soogil Hong**: Formal analysis; Investigation; Visualization; Writing—review and editing. **Hasan F Alnaser**: Formal analysis; Investigation; Writing—review and editing. **Eleni P Mimitou**: Resources; Investigation; Methodology; Writing—review and editing. **Keun P Kim**: Conceptualization; Formal analysis; Supervision; Funding acquisition; Investigation; Visualization; Writing—review and editing. **Hajime Murakami**: Conceptualization; Formal analysis; Supervision; Funding acquisition; Investigation; Visualization; Writing—original draft; Writing—review and editing. **Scott Keeney**: Conceptualization; Formal analysis; Supervision; Funding acquisition; Visualization; Writing—original draft; Writing—review and editing.

Source data underlying figure panels in this paper may have individual authorship assigned. Where available, figure panel/source data authorship is listed in the following database record: biostudies:S-SCDT-10_1038-S44318-024-00318-8.

## Disclosure and competing interests statement

The authors declare no competing interests.

# Expanded View Figures

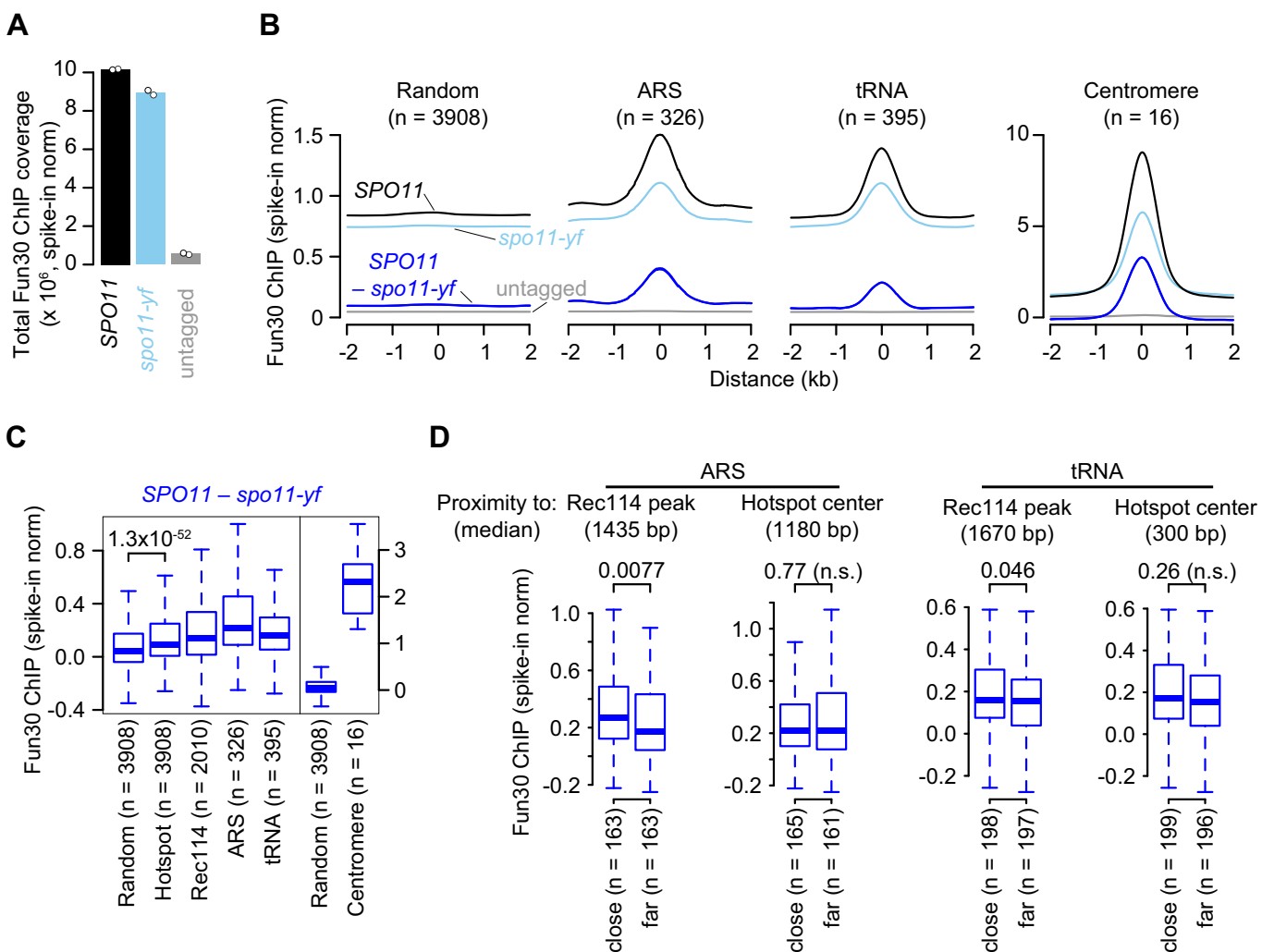

**Figure EV1. DSB-dependent Fun30 enrichment.**

(A) Total Fun30 ChIP-seq coverage normalized to the spike-in control. Bars are the means from two biological replicates; open circles show the individual values for each replicate. (B) Average Fun30 ChIP-seq signals around ARS, tRNA, and centromere. The random sites here and in (C) are the same as in Fig. 3B. (C) DSB-dependent Fun30 enrichment. Box plots summarize the distributions across all of the indicated elements from (B) and Fig. 3B for Fun30 ChIP-seq signal summed in 1-kb windows. Note the different y-axis scales for left and right parts of the plot. In all box plots, thick horizontal bars denote medians, box edges mark the upper and lower quartiles, and whiskers indicate values within 1.5-fold of the interquartile range. Outliers are not shown. Here and in (D), numbers above brackets indicate *p* values of two-sided Wilcoxon tests. (D) ARS and tRNAs that are closer to Rec114 binding sites tend to exhibit higher DSB-dependent Fun30 ChIP-seq signals. The ARS and tRNA regions from (B, C) were subdivided into two groups based on the distance to the nearest Rec114 peak or hotspot center: "close" indicates elements less than the median distance away and "far" indicates the rest. Proximity to Rec114 peaks was associated with a significantly higher DSB-dependent Fun30 ChIP-seq signal, whereas proximity to hotspots showed no such pattern. These results suggest that at least some of the DSB-dependent recruitment of Fun30 to ARS or tRNA genes is a consequence of fortuitous proximity or overlap of (some of) these elements with Rec114 ChIP peaks.

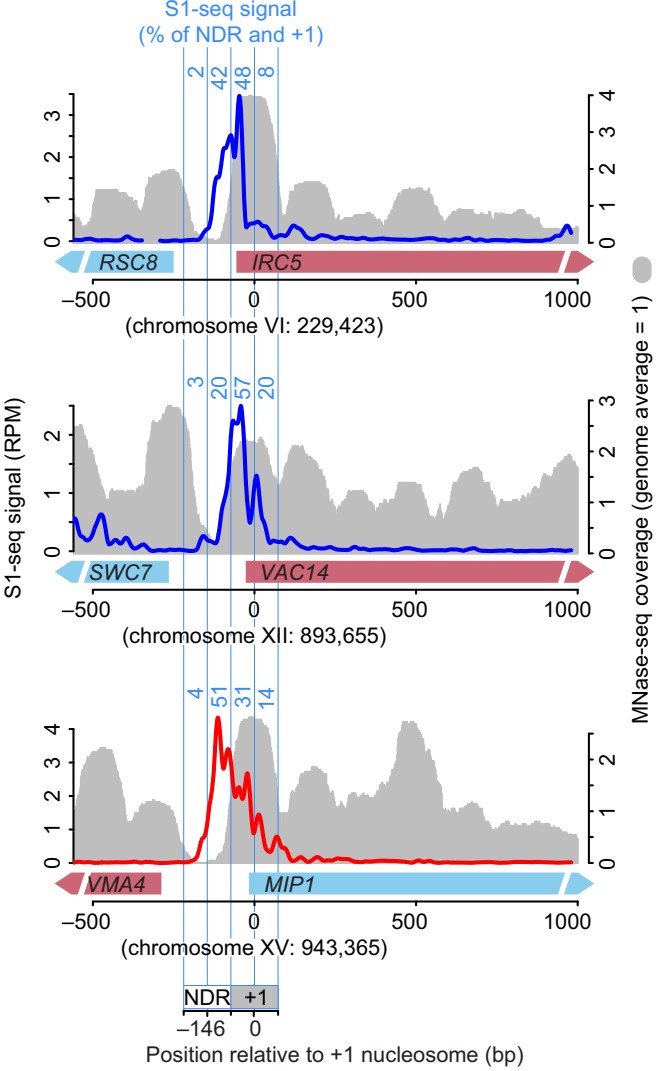

**Figure EV2. MRX/Sae2 nicks within the +1 nucleosome.**

Examples of resection endpoint distributions in *exo1-nd fun30Δ* at three representative loci that contributed to the average shown in Fig. 4B. S1-seq signals (41-bp smoothed) from the top (blue) or bottom (red) strand are shown, dependent on the orientation of the gene where the +1 nucleosome is located. Numbers in light blue indicate the percentages of S1-seq signal in the four windows spanning the +1 nucleosome and NDR (see legend to Fig. 4B).

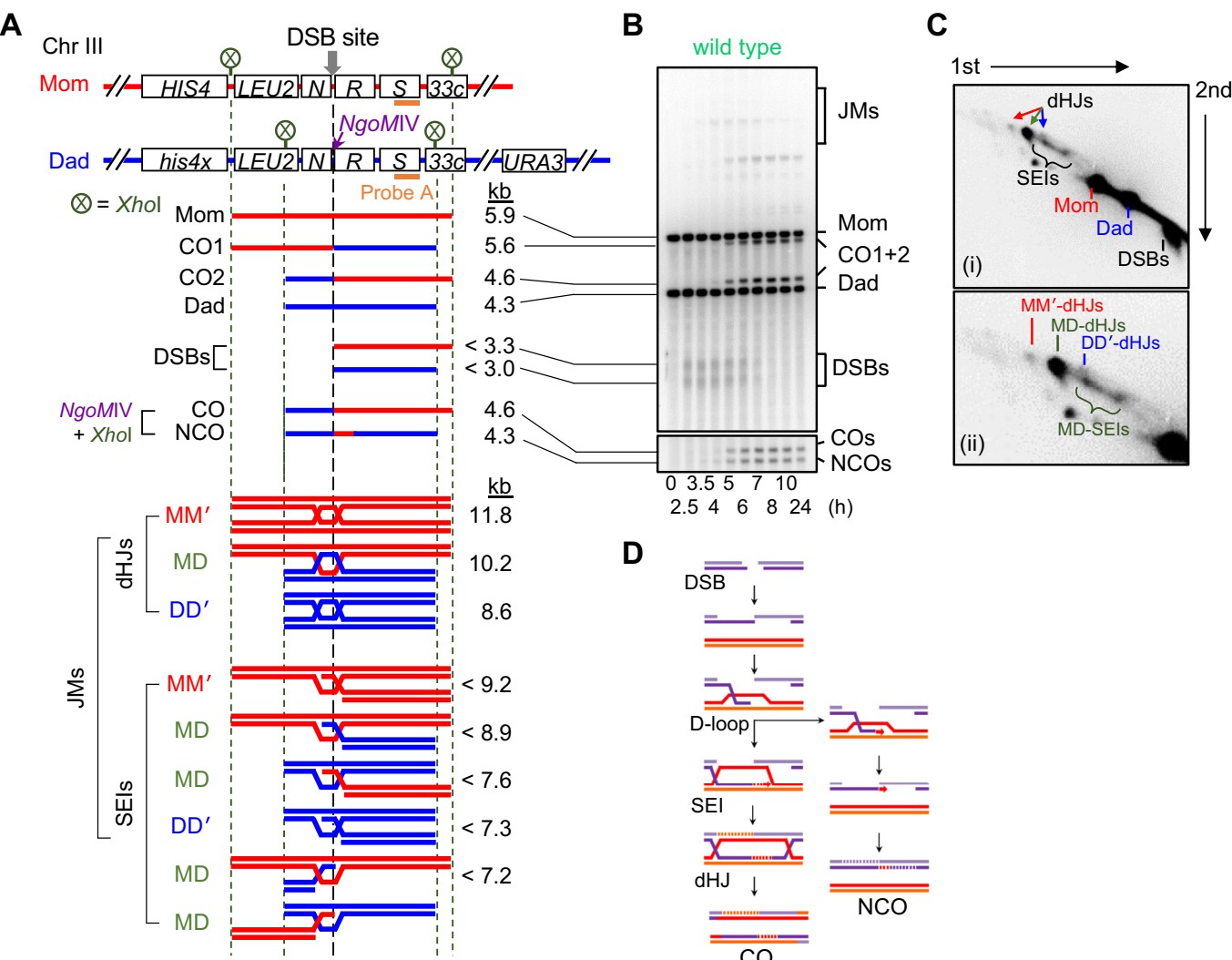

**Figure EV3.  Physical assay detecting recombination intermediates at the *HIS4LEU2* hotspot.**

(A) Physical map of the *HIS4LEU2* locus showing diagnostic *Xho*I restriction enzyme sites and the position of Southern blot probe A. "Mom" and "Dad" indicate the two parental versions of the locus; COs, crossovers; NCOs, noncrossovers; MM' IS-dHJ, intersister double-Holliday junction; MD IH-dHJ, interhomolog double-Holliday junction; DD' IS-dHJ, intersister double-Holliday junction; SEIs, single-end invasions. Positions of *Xho*I sites are indicated as circled Xs. (B) Example one-dimensional gel analysis showing parental signals, DSBs, COs, NCOs, and joint molecules (JMs). The Southern blot images are reproduced from Fig. 6A,F. (C) Example two-dimensional gel displaying parental signals and recombination intermediates. Green arrow or text indicate interhomolog species; red and blue arrows and text indicate intersister species. (D) Key steps in crossover and noncrossover formation during meiosis.

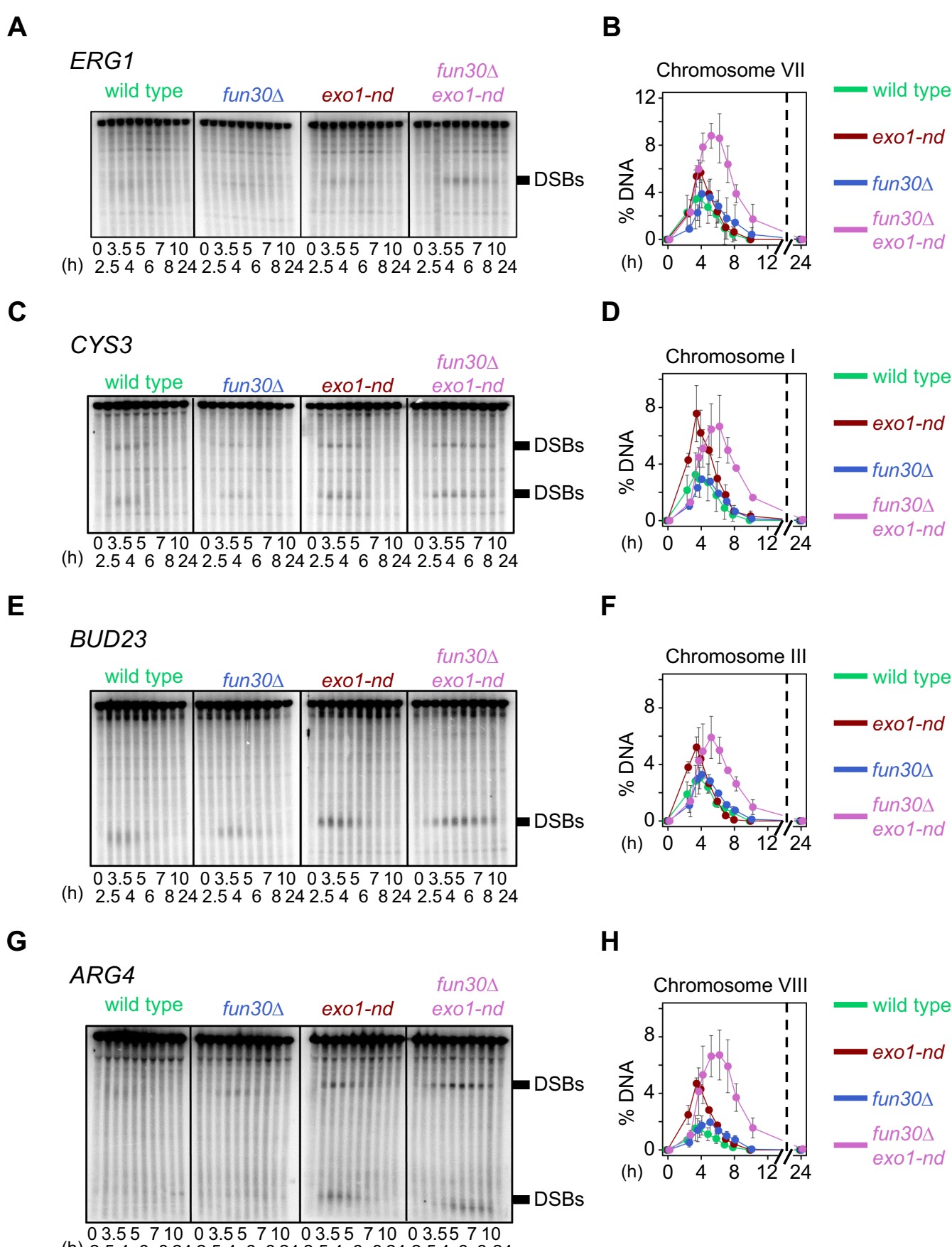

◀ **Figure EV4. DSBs formation at various natural hotspots.**

(**A–H**) Representative one-dimensional gel analyses of DSBs and corresponding quantification at the *ERG1* (**A**, **B**), *CYS3* (**C**, **D**), *BUD23* (**E**, **F**), and *ARG4* (**G**, **H**) hotspots. Error bars indicate mean ± SD for three independent cultures.

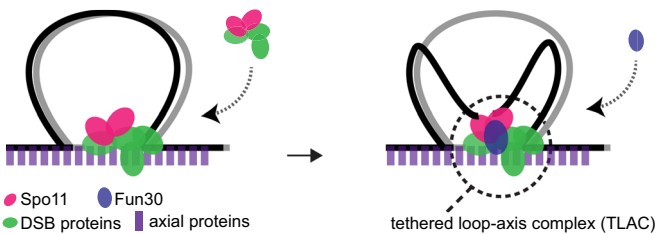

**Figure EV5.   Schematic representation of DSB-dependent Fun30 recruitment in the tethered loop-axis complex.**

A model proposed based on findings in this study. In response to Spo11 cleaving DNA within the TLAC, Fun30 is recruited to the DSB ends and remodels the nucleosomes.

