## [Peer Review File · The EMBO Journal]

Meiotic DNA break resection and recombination rely on chromatin remodeler Fun30

Pei-Ching Huang, Soogil Hong, Hasan Alnaser, Eleni Mimitou, Keun Kim, Hajime Murakami, and Scott Keeney

Corresponding author(s): Scott Keeney (s-keeney@ski.mskcc.org) , Hajime Murakami (hajime.murakami1@abdn.ac.uk)

Review Timeline:

Submission Date:	30th Apr 24
Editorial Decision:	31st May 24
Revision Received:	3rd Oct 24
Accepted:	12th Nov 24

Editor: Hartmut Vodermaier / Ioannis Papaioannou

Transaction Report:

Dear Dr. Keeney,

Thank you for submitting your manuscript EMBOJ-2024-117736 for consideration by The EMBO Journal. It has now been seen by three experts in the field, and we have received the full set of their comments, which you can find below.

As you will see, all referees recognize the high quality of the presented data as well as the novelty and significance of the findings. They also list a number of constructive suggestions for further improvement of the study and the manuscript, which we would kindly ask you to address in a revised version.

Given the referees' comments and recommendations, I would like to invite you to submit a revised version of the manuscript along with a detailed point-by-point response addressing all referees' comments. I should add that it is EMBO Journal policy to allow only a single round of major revision, and acceptance of your manuscript will therefore depend on the completeness of your responses in this revised version. Please let me know if you have any questions or comments that you would like to discuss with me.

We generally allow three months as standard revision time (August 30, 2024). As a matter of policy, competing manuscripts published during this period will not negatively impact our assessment of the conceptual advance presented by your study. However, we request that you contact us as soon as possible upon publication of any related work, to discuss how to proceed. Should you foresee a problem in meeting this three-month deadline, please let us know in advance and we may be able to grant an extension.

Thank you for the opportunity to consider your work for publication in The EMBO Journal. I look forward to your revision.

Best regards,

Ioannis

Instructions for preparing your revised manuscript

1. When you are ready to submit the revision, please upload:

- A Word file of the manuscript text (including legends of main Figures, EV Figures and Tables). Please make sure that changes are highlighted (or "tracked") to be clearly visible.

- Individual production-quality figure files (one file per figure). When assembling your figures, please refer to our figure preparation guidelines in order to ensure proper formatting and readability in print as well as on screen:

If the data shown in a figure are obtained from n {less than or equal to} 2, please use scatter plots showing the individual data points.

- i. the name of the statistical test used to generate error bars and P values
- ii. the number (n) of independent experiments (please specify technical or biological replicates) underlying each data point (discussion of statistical methodology can be reported in the Materials and Methods section, but figure legends should contain a basic description of n , P , and the test applied)
- iii. the nature of the bars and error bars (s.d., s.e.m.).

- A point-by-point response to the referees' comments, with a detailed description of the changes made (as a word file). All referees' concerns must be fully addressed and their suggestions taken on board. When preparing your letter of response to the referees' comments, please bear in mind that this will form part of the Review Process File and will therefore be available online to the community. Please note that you have the possibility to opt out of the transparent process at any stage prior to publication by letting the editorial office know (contact@embojournal.org); if you do opt out, the Review Process File link will point to the

following statement: "No Review Process File is available with this article, as the authors have chosen not to make the review process public in this case.". For more details on our Transparent Editorial Process, please visit our website: <https://www.embopress.org/page/journal/14602075/authorguide#transparentprocess>

- Expanded View (EV) files (replacing Supplementary Information) that are collapsible/expandable online. A maximum of 5 EV Figures can be typeset. EV Figures should be cited as "Figure EV1, Figure EV2" etc. in the text, and their respective legends should be included in the manuscript file after the legends of regular figures. See detailed instructions regarding Expanded View files here:

- For the figures that you do NOT wish to display as Expanded View figures, they should be bundled together with their legends in a single PDF file called "Appendix", which should start with a short Table of Contents (including page numbers). Appendix figures should be referred to in the main text as: "Appendix Figure S1, Appendix Figure S2" etc. Please see detailed instructions here: <https://www.embopress.org/page/journal/14602075/authorguide#expandedview>

- A complete author checklist, which you can download from our author guidelines (<https://www.embopress.org/page/journal/14602075/authorguide>). Please note that the checklist will also be part of the Review Process File.

2. Please note that no statistics should be calculated and shown in Figures if $n=2$. Please also note that each p value should be reported as an exact value.

3. Before submitting your revision, primary datasets (and computer code, where appropriate) produced in this study need to be deposited in appropriate public databases (see <https://www.embopress.org/page/journal/14602075/authorguide#dataavailability>).

In particular, you are kindly requested to make sure that all S1-seq and ChIP-seq sequencing data produced in your study are deposited in appropriate databases. The accession numbers and databases should be listed in a formal "Data availability" section (placed after Materials and Methods) that follows the model below (see also <https://www.embopress.org/page/journal/14602075/authorguide#dataavailability>):

Data availability

- RNA-seq data: Gene Expression Omnibus GSE46843 (<https://www.ncbi.nlm.nih.gov/geo/query/acc.cgi?acc=GSE46843>)
- [data type]: [name of the resource] [accession number/identifier/doi] ([URL or identifiers.org/DATABASE:ACCESSION])

*** All links should resolve to a page where the data can be accessed. ***

*** Please remember to provide in the Data availability section of your revised manuscript reviewer passwords if the datasets are not yet public. ***

*** The Data Availability Section is restricted to new primary data that are part of this study. In case you have no data that require deposition in a public database, please state so instead of referring to the database: "Our study includes no data deposited in public repositories." under the heading "Data availability". ***

4. Please check that the title and the abstract of the manuscript are brief, yet explicit, even to non-specialists. The length of the title should not exceed 100 characters, and the abstract should be a single paragraph not exceeding 175 words.

5. Please also note our reference format: <https://www.embopress.org/page/journal/14602075/authorguide#referencesformat>.

7. Please remember: digital image enhancement is acceptable practice, as long as it accurately represents the original data and conforms to community standards. If a figure has been subjected to significant electronic manipulation, this must be noted in the figure legend or in the "Materials and Methods" section. The editors reserve the right to request original versions of figures and the original images that were used to assemble the figure.

8. Our journal encourages inclusion of data citations in the reference list to directly cite datasets that were obtained from public databases. Data citations in the article text are distinct from normal bibliographical citations and should directly link to the database records from which the data can be accessed. In the main text, data citations are formatted as follows: "Data ref: Smith et al, 2001" or "Data ref: NCBI Sequence Read Archive PRJNA342805, 2017". In the Reference list, data citations must

be labeled with "[DATASET]". A data reference must provide the database name, accession number/identifiers, and a resolvable link to the landing page from which the data can be accessed at the end of the reference. Further instructions are available at: <https://www.embopress.org/page/journal/14602075/authorguide#referencesformat>.

9. We request authors to consider both actual and perceived competing interests. Please review our policy (<https://www.embopress.org/page/journal/14602075/authorguide#conflictsofinterest>) and update your competing interests statement if necessary. Please name this section 'Disclosure and competing interests statement' and place it after the Acknowledgements section.

10. Please note that all corresponding authors are required to provide an ORCID ID upon submission of a revised manuscript (<https://orcid.org/>). Please find instructions on how to link your ORCID ID to your account in our manuscript tracking system in our Author guidelines (<https://www.embopress.org/page/journal/14602075/authorguide#authorshipguidelines>).

11. We use CRediT to specify the contributions of each author in the journal submission system. CRediT replaces the author contribution section, which should be removed from the manuscript. Please use the free text box to provide more detailed descriptions. See also guide to authors: <https://www.embopress.org/page/journal/14602075/authorguide#authorshipguidelines>.

13. We would also welcome the submission of cover suggestions or motifs to be used by our Graphics Illustrator in designing a cover.

14. Please use the link below to submit your revision:
Link Unavailable

Referee #1:

In the manuscript "Meiotic DNA break resection and recombination rely on chromatin remodeller Fun30" Pei-Ching Huang and colleagues investigate the involvement of ATP-dependent nucleosome remodellers in meiotic recombination in the budding yeast *Saccharomyces cerevisiae*. In their previous work, the Keeney lab had shown an influence of nucleosomes, the basic unit of chromatin, on the process of meiotic DSB resection. Consistently, previous biochemical work had shown that nucleosomes can form a barrier to the resection nuclease Exo1 and previous cell biological work on the resection of DSBs in mitotic cell cycles had shown that resection is coupled to nucleosome eviction. Lastly, several nucleosome remodellers including Fun30 had been implicated in the process. This investigation focusses on the role of nucleosome remodellers in meiotic recombination and starts with a candidate screen of several nucleosome remodeller mutants, in which a *fun30* strain showed a strong defect in meiotic DSB resection. Epistasis analysis with resection nuclease mutants, as well as analysis of resection tracts from S1-seq data suggest an involvement of Fun30 in both short-range resection by the Mre11-complex as well as long-range resection by Exo1. Consistently, Fun30 is found to localize to meiotic DSBs. The *fun30 exo1-nd* mutant in particular has very short resection tracks, which translate into persistent DSBs, reduced formation of cross-overs, a reduction in interhomolog, increased meiosis I non-disjunctions and a reduced spore viability. Altogether, these data point towards a crucial role of Fun30 in meiotic DSB processing and recombination.

This is an impressive manuscript. All the experimental data are very clear and of high quality and the manuscript is very well written, making the work accessible to the broad audience of The EMBO Journal. I have only few points, which I would like to see address prior to publication.

Major points:

#1 - the authors don't give details on biological replication for many of their experiments. While they generally observe very large effects, which are very likely to be reproducible over biological replicates in their system, this information will nonetheless be useful to the reader.

#2 - from the data in Fig. 1 and 2 the authors conclude that Fun30 supports resection by both the Mre11-complex and Exo1. The genetic data gives sufficient support for Fun30 promoting resection by the Mre11-complex. However, the support for an Exo1 supporting role is not as strong. It mainly comes from the comparison of resection between WT and *fun30* strains together with the idea that differences in resection >500 bp away has to be due to Exo1-dependent resection. It would be good if this point could be clarified in the discussion.

#3 - in Fig. 4 the authors overlaid resection endpoints from their S1-seq data of WT, *fun30*, *exo1-nd* and *fun30 exo1-nd* strains with a meiotic MNase-seq dataset (Pan et al. 2011). The weakness of this analysis is that it does not account for chromatin dynamics. Specifically, the nucleosome positioning is likely going to be different in different remodeller mutants and it will likely

also depend on resection. The authors acknowledge these limitations in the discussion and I agree that it would be beyond the scope of the current study to map nucleosome position in all conditions. However, I think the conclusions from this figure need to be toned down. Also, the authors should think of a way to indicate that the nucleosome positioning shown in the individual figure panels is from WT cells and not from the respective mutant.

Minor points:

- Is Fig. 3A a good representative of their Fun30 ChIP data? In this window, it looks like there is association of Fun30 with at least two hot-spots in the *spo11-yf* mutant, but this seems not to be a general trend.
- Information given in figure legends is often limited, e.g. blue (*fun30* ?) and purple (*fun30 exo1-nd*?) curves in Fig. 4D are not explained.
- Also, the blue curves in Fig. 4C and D are different. I presume this is because the clustering has been done based on the *fun30 exo1-nd* dataset, but additional explanation would be helpful.
- Fig. 5D needs additional explanation for statistics/error bars in the figure legend.

Referee #2:

The Exo1 exonuclease is a component of a two-step mechanism that acts to resect meiotically induced double-strand breaks (DSBs) catalyzed by Spo11. Its activity is strongly blocked by nucleosomes *in vitro*. These observations and previous studies examining the role of the Fun30 chromatin remodeler in promoting resection of DSBs in vegetative cells encouraged Huang et al. to test if a chromatin remodeling activity is required to resect meiotically induced DSBs present in/near nucleosomal DNA. Using a set of extremely elegant genetic (testing the effects of different remodelers on resection), genomic (mapping resection tracks with respect to meiotic DSBs and nucleosome positioning) and molecular (mapping Fun30 localization sites with respect to meiotic DSBs and Rec114 positioning; measuring meiotic recombination intermediates in mutant backgrounds), methods the authors determined that:

1. Fun30 plays a major role in meiotic DSB break resection; I agree with their conclusion that "Fun30 directly promotes both the MRX- and Exo1-dependent steps in resection, possibly by removing nucleosomes from broken chromatids."
2. Interhomolog recombination bias is significantly reduced in *fun30 exo1-nd* mutants which are severely compromised for resection. This observation explains why a critical resection length is needed for recombination that promotes chromosome segregation in Meiosis I.

This is a beautiful study that uses sophisticated methods to make a compelling case for Fun30 playing a major role in meiotic DSB break resection. While it would have been interesting to do experiments that provide a more detailed explanation for the loss of interhomolog bias in *fun30 exo1-nd* mutants, such work should be left for another day. I have only minor comments.

Comments

1. Line 97, line 131. Figure 2. The analyses presented in these parts of the paper assume that resection in *exo1-nd* is completely abrogated. It's worth noting that for nucleases that act through a two-metal catalysis mechanism (like Exo1) altering a single metal binding residue may not fully ablate function. Lee et al. (NAR 30:942) showed that the human *exo1-D78A* and *exo1-D173A* mutant proteins display nuclease activities, though at levels significantly lower than the wild-type protein. I think that it would be useful to include this caveat.
2. Line 286. The authors state that the *exo1-nd* mutation increased both the time averaged amount of DSB signal and recombination intermediates but did not affect crossover levels. The DSB levels for *exo1-nd*, as measured by % DNA in Figure 6B, appear different from those seen for *exo1-nd* by Zakharyevich et al. (Figure 4D, Mol Cell 40:1001; note that both studies looked DSB levels in *HIS4LEU2*). Do the authors have an explanation for this difference? It's curious because both studies show similar kinetics for crossover formation. At the very least it's worth pointing out this difference.
3. It seems valuable to have the Figure S7 summary image in the main text. If space is an issue perhaps include it as the last panel in Figure 7?
4. Line 501. It would be helpful to provide a bit more detail on how the sporulation protocol was performed. Specifically, what was the exact dilution for the YPD-cultured cells (line 501)? What was the volume of the sporulation media (line 504)?
5. Line 611. It would be useful to provide a reference for the custom reference genome used. Was it S288c derived? SK1 derived? If it was S288c, it would be useful to reference the previous methods that account d for the polymorphism/genome differences between the two strains backgrounds.
6. Figure 1. Please indicate if these blots are representative of X repeats.
7. Figure 4D. It would be useful to provide a bit more explanation of what the purple and blue peaks represent.

8. Suggested word change, line 70: Fun30, a SWI-SNF-like ATPase that promotes resection for long distances.
9. Add a word, line 80: ...endonuclease to remove ssDNA and then visualized....
10. References appear incomplete. For example see: Cannavo and Cejka; Costelloe et al.; Hu et al.; Hunter; Mimitou et al.; Neves-Costa et al.; Smith and Roeder; Zakharyevich et al. 2010.

Referee #3:

DNA double-strand break resection initiates the homologous recombination pathway. Resection takes place in the context of chromatin, and nucleosomes have been proposed to hinder this reaction by Exo1. Here, Huang et al. investigated the role of the Swi2/Snf2 chromatin remodeler Fun30 in resection of meiotic DNA breaks, DNA joint molecules metabolism and spore viability, individually or in combination with an *exo1* mutant partly defective for DNA break resection. Using high-throughput molecular assays, they show that Fun30 is recruited to DSB hotspots in a manner that depends on the catalytic activity of Spo11, and that it promotes DNA end resection genome-wide. Through careful analysis of the resection tract length in the *fun30D*, *exo1-nd* and the double-mutant, as well as through correlation with the nucleosome occupancy map around DSB hotspots, they deduce that Fun30 promotes both resection initiation and resection extension by remodeling nucleosomes. They rule out similar roles of various other chromatin remodelers. Finally, they analyze the absolute levels of DSBs, of two types of crossover-designated DNA joint molecules (SEI and dHJs), and of non-crossover and crossover products over meiotic time courses in these single and double mutants at two strong DSB hotspots. This analysis reveals recombination defects specific to the *fun30* *exo1-nd* double-mutant, for both homolog bias and product formation, accompanied by reduced spore viability. They discuss these downstream defects in light of the severe resection defect of the double-mutant.

The experiments and analysis are of very high quality, presented in a clear and logical order, and support the conclusions of the paper. They extend to meiosis the resection-promoting function of Fun30 identified in mitotic cells, which is a worthy addition to the field. The analysis of the downstream consequences on the recombination reaction leads the author to postulate functions of resection beyond simply generating enough ssDNA for Rad51-Dmc1 filament assembly. This latter part leads to a more speculative discussion. Overall I only have a few, easily addressable comments, which I feel could strengthen the work and enrich parts of the discussion.

1. The role of the catalytic-deficient mutant Fun30 is not investigated, only the deletion mutant. Surprising catalytic-independent functions have been observed in meiosis for proteins that otherwise mainly assumed resection functions in mitosis, such as Exo1. Since this is the first time that the role of Fun30 is carefully examined in meiosis, ascertaining the dependency of the resection phenotype and downstream functions on its catalytic activity would be a worthwhile addition to the manuscript.
2. The double mutant *exo1-nd fun30D* exhibits increased DSB, IS-dHJ and IH-dHJ relative to WT or any of the single mutants. Yet it fails to produce both inter-homolog CO and NCO (Fig. S4F), suggesting that the disappearance of the abundant IH DNA joint molecules between 8 and 12h mainly reflects a failure to repair off the homolog. Plotting these different data alongside each other would be useful. Furthermore, the CO-designated SEI intermediates are not different between *exo1-nd* and *exo1-nd fun30D* (Fig 6E), while CO production is. It suggests that, in a context in which resection is strongly limited, or in which *exo1* is catalytically inactive, Fun30 becomes critical for CO maturation. This point may be worth discussing.
3. Along this line, a refined analysis of SEI data presented in Figure 6 could yield greater insights into the mechanism by which Exo1 (and to a lesser extent Fun30) promotes homolog bias. Indeed, while the loss of the inter-homolog bias in the *fun30* *exo1-nd* double mutant is very obvious at the dHJ level, this does not seem to be the case at the SEI level (gels Fig 6D). The SEI decomposition as IS and IH is not presented in Fig 6E, unlike for dHJ. If the defect in homolog bias indeed occurs at the SEI to dHJ transition, it may suggest a defect in reuniting the two ends specifically on the homolog, ruling out defects in identifying the homolog, or in failing to suppress usage of the sister. Instead it would point at a defect in annealing the second end, which could be more easily overcome upon spatial tethering of the DSB and the donor (i.e. the sister). Such activity may require longer ssDNA, or be promoted by the role of Exo1 catalytic activity in DSB end-tethering in mitotic cells (Nakai .. Resnick DNA Repair 2011; Piazza .. Koszul NCB 2021), like other resection mutants. Although investigating the role of these proteins in DSB end-tethering in meiosis is out of the scope of this study, it could be mentioned in the relevant discussion section. In line with the interpretation that end-tethering may be compromised, both *exo1-nd* and *exo1-nd fun30* mutants exhibit reduced spore viability that cannot be explained solely by non-disjunction death. Instead, they could result from unrepaired, detached and randomly segregating chromosome fragments.
4. A previous work from the team showed that S1-seq allows to detect ssDNA of opposite polarity to that expected for resection, that depended on Dmc1 (Mimitou 2017). This signal was interpreted as reflecting the presence of D-loop DNA joint molecules at the donor site. Analysis of this signal here may yield insights as to the individual and combined role of Fun30 and Exo1 catalytic function in D-loop formation and/or transition to intermediates detectable by 2D-gel (SEI, dHJs), and further substantiates the discussion about putative homology search defects.

Minor comments:

- Genetic distance in Fig 5D would be more informative if data from the single *exo1-nd* and *fun30* mutants were shown alongside the double mutant.
- Fig 6E: The sum of dHJ does not seem to correspond to the sum of IH- and IS-dHJ measured, particularly for the double mutant.
- The *exo1-nd* values are inconsistent between what is shown in plot in Fig 7C and 7D: IS/IH is ~1 in 7C and 0.25-0.5 in 7D.
- The S1-seq protocol appears to have been significantly improved relative to the previously published version. It may be of interest to the community to have a full updated protocol in supplementary materials in addition to the long list of modifications in the main methods.

Response to reviews

We thank the referees for their detailed and constructive comments. We are pleased to have been given the opportunity to respond in the form of a revised manuscript. We have addressed all of the points raised by inclusion of new data and analyses, revisions to text or figures, and/or detailed responses below. We feel the paper has been substantially improved as a result. Below, referee comments are in black text, responses are in blue.

Overview of major changes:

1. Inclusion of single mutants in measuring genetic distance and chromosome missegregation (**Figure 5D, Appendix Table S2 and S3**). An author was added (H.F. Alnaser) because of collection of new data; all authors agreed to the addition and the updated author order.
2. Rewriting to clarify, discuss interpretations and limitations more thoroughly, and meet *EMBO Journal* formatting and length requirements. Line numbers refer to the clean Word document without tracked changes; depending on how changes are displayed, the line numbering may be different in the tracked-changes version.

Referee #1:

In the manuscript "Meiotic DNA break resection and recombination rely on chromatin remodeller Fun30" Pei-Ching Huang and colleagues investigate the involvement of ATP-dependent nucleosome remodellers in meiotic recombination in the budding yeast *Saccharomyces cerevisiae*. In their previous work, the Keeney lab had shown an influence of nucleosomes, the basic unit of chromatin, on the process of meiotic DSB resection. Consistently, previous biochemical work had shown that nucleosomes can form a barrier to the resection nuclease Exo1 and previous cell biological work on the resection of DSBs in mitotic cell cycles had shown that resection is coupled to nucleosome eviction. Lastly, several nucleosome remodellers including Fun30 had been implicated in the process. This investigation focusses on the role of nucleosome remodellers in meiotic recombination and starts with a candidate screen of several nucleosome remodeller mutants, in which a *fun30Δ* strain showed a strong defect in meiotic DSB resection. Epistasis analysis with resection nuclease mutants, as well as analysis of resection tracts from S1-seq data suggest an involvement of Fun30 in both short-range resection by the Mre11-complex as well as long-range resection by Exo1. Consistently, Fun30 is found to localize to meiotic DSBs. The *fun30Δ exo1-nd* mutant in particular has very short resection tracks, which translate into persistent DSBs, reduced formation of cross-overs, a reduction in interhomolog, increased meiosis I non-disjunctions and a reduced spore viability. Altogether, these data point towards a crucial role of Fun30 in meiotic DSB processing and recombination.

This is an impressive manuscript. All the experimental data are very clear and of high quality and the manuscript is very well written, making the work accessible to the broad audience of The EMBO Journal. I have only few points, which I would like to see address prior to publication.

We thank the reviewer for the positive response and the constructive suggestions.

Major points:

#1 - the authors don't give details on biological replication for many of their experiments. While they generally observe very large effects, which are very likely to be reproducible over biological replicates in their system, this information will nonetheless be useful to the reader. We added information about the number of biological replicates in figure legends.

#2 - from the data in Fig, 1 and 2 the authors conclude that Fun30 supports resection by both the Mre11-complex and Exo1. The genetic data gives sufficient support for Fun30

promoting resection by the Mre11-complex. However, the support for an Exo1 supporting role is not as strong. It mainly comes from the comparison of resection between WT and *fun30Δ* strains together with the idea that differences in resection >500 bp away has to be due to Exo1-dependent resection. It would be good if this point could be clarified in the discussion.

We revised the text to make the logic clearer (P7, L140–141). We certainly agree that the evidence for a role for Fun30 in promoting Exo1 activity is less direct than for MRX, but we think the argument is a logically robust one nonetheless. For our inference to be incorrect, it would have to be the case that a) MRX resects farther when Exo1 is active than when Exo1 is inactive, but b) only does so in the presence of Fun30. Both conditions are difficult to envision, so the combined scenario appears to us to be highly implausible.

#3 - in Fig. 4 the authors overlaid resection endpoints from their S1-seq data of WT, *fun30Δ*, *exo1-nd* and *fun30Δ exo1-nd* strains with a meiotic MNase-seq dataset (Pan et al. 2011). The weakness of this analysis is that it does not account for chromatin dynamics. Specifically, the nucleosome positioning is likely going to be different in different remodeler mutants and it will likely also depend on resection. The authors acknowledge these limitations in the discussion and I agree that it would be beyond the scope of the current study to map nucleosome position in all conditions. However, I think the conclusions from this figure need to be toned down. Also, the authors should think of a way to indicate that the nucleosome positioning shown in the individual figure panels is from WT cells and not from the respective mutant.

We added explanation in the text (P10 L208) and reinforced in the **Fig. 4** legend that the MNase-seq map is from wild type, and we added a paragraph further explaining limitations of our study (P19 L411–421). A fundamental problem we face is that even at a strong DSB hotspot, most of the chromatids in a population of cells are not broken. Thus, MNase-seq and related bulk population methods struggle to detect what is happening on broken chromatids against a much larger background of unbroken ones.

The reviewer mentions the issue that we do not know if nucleosome positioning is different in chromatin remodeler mutants. This is true, but it is not clear that it would be likely to change the conclusions from the analysis in **Fig. 4C,D**. Moreover, we show directly that absence of Fun30 has no detectable effect on the distribution of DSBs at hotspots (**Fig. 1B.iv and Appendix Fig. S1C**). It is well established that Spo11 is highly sensitive to chromatin structure, so it appears that absence of Fun30 has fairly modest effects on bulk chromatin structure on unbroken chromatids at the genomic locations that we are examining in **Fig. 4**. This conclusion is also consistent with the mild effects of *fun30Δ* on chromatin structure seen in other contexts (Byeon et al, 2013).

Minor points:

- Is Fig. 3A a good representative of their Fun30 ChIP data? In this window, it looks like there is association of Fun30 with at least two hot-spots in the *spo11-yf* mutant, but this seems not to be a general trend.

We thank the reviewer for pointing this out. In fact, the two Fun30 ChIP peaks in the *spo11-yf* mutant noted by the reviewer do not match the peaks of Spo11 oligos. To assist readers in making comparisons, we added finer tick marks on the x-axis scale and duplicated the scale in the upper graph.

- Information given in figure legends is often limited, e.g. blue (*fun30Δ*?) and purple (*fun30Δ exo1-nd*?) curves in Fig. 4D are not explained.

Thank you. We used the same color coding for the mutants throughout the figure (and indeed throughout the paper), but only explicitly indicated which color is which mutant in **Fig. 4A**. We recognize that this can be confusing, so we added info to the legend to clarify.

- Also, the blue curves in Fig. 4C and D are different. I presume this is because the

clustering has been done based on the fun30 Δ exo1-nd dataset, but additional explanation would be helpful.

The reviewer is correct. We added a sentence to the figure legend to explain.

- Fig. 5D needs additional explanation for statistics/error bars in the figure legend.

Figure legend updated.

Referee #2:

The Exo1 exonuclease is a component of a two-step mechanism that acts to resect meiotically induced double-strand breaks (DSBs) catalyzed by Spo11. Its activity is strongly blocked by nucleosomes *in vitro*. These observations and previous studies examining the role of the Fun30 chromatin remodeler in promoting resection of DSBs in vegetative cells encouraged Huang et al. to test if a chromatin remodeling activity is required to resect meiotically induced DSBs present in/near nucleosomal DNA. Using a set of extremely elegant genetic (testing the effects of different remodelers on resection), genomic (mapping resection tracks with respect to meiotic DSBs and nucleosome positioning) and molecular (mapping Fun30 localization sites with respect to meiotic DSBs and Rec114 positioning; measuring meiotic recombination intermediates in mutant backgrounds), methods the authors determined that:

1. Fun30 plays a major role in meiotic DSB break resection; I agree with their conclusion that "Fun30 directly promotes both the MRX- and Exo1-dependent steps in resection, possibly by removing nucleosomes from broken chromatids."
2. Interhomolog recombination bias is significantly reduced in *fun30 exo1-nd* mutants which are severely compromised for resection. This observation explains why a critical resection length is needed for recombination that promotes chromosome segregation in Meiosis I.

This is a beautiful study that uses sophisticated methods to make a compelling case for Fun30 playing a major role in meiotic DSB break resection. While it would have been interesting to do experiments that provide a more detailed explanation for the loss of interhomolog bias in *fun30 exo1-nd* mutants, such work should be left for another day. I have only minor comments.

Thank you for the positive feedback and constructive comments.

Comments

1. Line 97, line 131. Figure 2. The analyses presented in these parts of the paper assume that resection in *exo1-nd* is completely abrogated. It's worth noting that for nucleases that act through a two-metal catalysis mechanism (like Exo1) altering a single metal binding residue may not fully ablate function. Lee et al. (NAR 30:942) showed that the human *exo1-D78A* and *exo1-D173A* mutant proteins display nuclease activities, though at levels significantly lower than the wild-type protein. I think that it would be useful to include this caveat.

Thank you for pointing this out. We agree that we cannot exclude that the mutant Exo1 protein contributes to resection. We do note, however, that the human mutant protein has greatly reduced activity (>55-fold to >650-fold reduction, depending on the substrate). Also, the resection phenotypes of the *exo1* point mutant and null mutant are indistinguishable from one another, both in yeast and in mouse, so it seems likely that any contribution of the mutant Exo1 protein is negligible for our purposes. We added a paragraph to address this caveat (P6, L103–110).

2. Line 286. The authors state that the *exo1-nd* mutation increased both the time averaged amount of DSB signal and recombination intermediates but did not affect crossover levels. The DSB levels for *exo1-nd*, as measured by % DNA in Figure 6B, appear different from those seen for *exo1-nd* by Zakharyevich et al. (Figure 4D, Mol Cell 40:1001; note that both studies looked DSB levels in *HIS4LEU2*). Do the authors have an explanation for this difference? It's curious because both studies show similar kinetics for crossover formation. At the very least it's worth pointing out this difference.

Measurements of DSBs and recombination intermediates are sensitive to multiple factors, including the synchrony of meiotic progression and detection procedures (e.g., sample preparation, Southern blotting, and signal quantification). As a result, achieving reproducible

outcomes can be challenging. Such variability is evident in Supplementary Figure S2A and B of the Zakharyevich publication (Zakharyevich et al, 2010), which showed two independent pairs of wild type and *exo1* Δ cultures. The two biological replicates showed strikingly opposite results for relative DSB timing and levels. As the reviewer points out, in Figure 4D of their publication, they did indeed observe similar DSB timing and levels in wild type and *exo1-nd*. We had also noticed the same point. However, Zakharyevich et al presented data from only a single replicate of *exo1-nd*, so there is no way to judge experimental variation. In contrast, we reproducibly observed higher DSB signals in *exo1-nd* compared to wild type at the *HIS4LEU2* locus in three biological replicates, as well as at four other (natural) hotspots. Because we are unable to draw any conclusions about similarities or differences given that their study had just a single biological replicate, and because this is not the primary focus of our study, we chose to present our findings without emphasizing these apparently differing conclusions.

3. It seems valuable to have the Figure S7 summary image in the main text. If space is an issue perhaps include it as the last panel in Figure 7?

Thank you for the suggestion. However, we consider this to be a somewhat subsidiary point in our story. Moreover, similar cartoons have been shown in several other papers by now, so there is no real conceptual advance conveyed by the figure itself. We therefore prefer to keep this panel supplemental, but we are taking a compromise position, which is to display this as an EV figure (**Fig. EV5**), which will be accessible in both the online (HTML) and downloadable (PDF) versions of the paper.

4. Line 501. It would be helpful to provide a bit more detail on how the sporulation protocol was performed. Specifically, what was the exact dilution for the YPD-cultured cells (line 501)? What was the volume of the sporulation media (line 504)?

We provided the requested details (P24, L526 and 530).

5. Line 611. It would be useful to provide a reference for the custom reference genome used. Was it S288c derived? SK1 derived? If it was S288c, it would be useful to reference the previous methods that account for the polymorphism/genome differences between the two strains backgrounds.

We apologize for the lack of clarity. We edited the methods section and added references (P29, L654–662). This was not really a custom genome per se, but rather simultaneous mapping against *sacCer2* (S288c-derived *S. cerevisiae* genome) and IFO1815 (*S. mikatae* genome). For purposes of mapping software input, these assemblies are combined into a single “genome” file. We continue to use *sacCer2* for mapping *S. cerevisiae* reads even though we are using SK1 samples so as to maintain consistency with the large amount of prior data our lab has generated, and because we previously showed that the polymorphisms between the biological sample and the reference genome have little to no effect on mapping ability (Pan et al, 2011).

6. Figure 1. Please indicate if these blots are representative of X repeats.

We indicated the number of replicates in the figure legend.

7. Figure 4D. It would be useful to provide a bit more explanation of what the purple and blue peaks represent.

Sorry for the lack of clarity. We clarified the legend for **Figures 4C and 4D**.

8. Suggested word change, line 70: Fun30, a SWI-SNF-like ATPase that promotes resection for long distances.

Changed as suggested.

9. Add a word, line 80: ...endonuclease to remove ssDNA and then visualized....

Changed as suggested.

10. References appear incomplete. For example see: Cannavo and Cejka; Costelloe et al.; Hu et al.; Hunter; Mimitou et al.; Neves-Costa et al.; Smith and Roeder; Zakharyevich et al. 2010.

Thank you. We updated all references.

Referee #3:

DNA double-strand break resection initiates the homologous recombination pathway. Resection takes place in the context of chromatin, and nucleosomes have been proposed to hinder this reaction by Exo1. Here, Huang et al. investigated the role of the Swi2/Snf2 chromatin remodeler Fun30 in resection of meiotic DNA breaks, DNA joint molecules metabolism and spore viability, individually or in combination with an *exo1* mutant partly defective for DNA break resection. Using high-throughput molecular assays, they show that Fun30 is recruited to DSB hotspots in a manner that depends on the catalytic activity of Spo11, and that it promotes DNA end resection genome-wide. Through careful analysis of the resection tract length in the *fun30Δ*, *exo1-nd* and the double-mutant, as well as through correlation with the nucleosome occupancy map around DSB hotspots, they deduce that Fun30 promotes both resection initiation and resection extension by remodeling nucleosomes. They rule out similar roles of various other chromatin remodelers. Finally, they analyze the absolute levels of DSBs, of two types of crossover-designated DNA joint molecules (SEI and dHJs), and of non-crossover and crossover products over meiotic time courses in these single and double mutants at two strong DSB hotspots. This analysis reveals recombination defects specific to the *fun30Δ* *exo1-nd* double-mutant, for both homolog bias and product formation, accompanied by reduced spore viability. They discuss these downstream defects in light of the severe resection defect of the double-mutant.

The experiments and analysis are of very high quality, presented in a clear and logical order, and support the conclusions of the paper. They extend to meiosis the resection-promoting function of Fun30 identified in mitotic cells, which is a worthy addition to the field. The analysis of the downstream consequences on the recombination reaction leads the author to postulate functions of resection beyond simply generating enough ssDNA for Rad51-Dmc1 filament assembly. This latter part leads to a more speculative discussion. Overall I only have a few, easily addressable comments, which I feel could strengthen the work and enrich parts of the discussion.

We appreciate the positive response and constructive suggestions.

1. The role of the catalytic-deficient mutant Fun30 is not investigated, only the deletion mutant. Surprising catalytic-independent functions have been observed in meiosis for proteins that otherwise mainly assumed resection functions in mitosis, such as Exo1. Since this is the first time that the role of Fun30 is carefully examined in meiosis, ascertaining the dependency of the resection phenotype and downstream functions on its catalytic activity would be a worthwhile addition to the manuscript.

We agree that this would be interesting to explore, and to this end we had already attempted to address this using a previously described ATPase catalytic-site mutant, *fun30-K603R* (Neves-Costa et al, 2009). This mutant exhibited a resection defect comparable to *fun30Δ* (Review Figure 1A). Unfortunately, however, the Fun30-K603R protein was unstable in vivo (Review Figure 1B), making these results inconclusive about the role of ATPase activity per se. Consequently, we prefer to exclude these data from the paper, but we can add them if the reviewer feels strongly about it.

Review Figure 1. The ATPase-deficient K603R mutation destabilizes Fun30 in vivo.

(A) Resection endpoint distributions detected by S1-Southern blotting at the *GAT1* hotspot in the indicated mutants.

(B) Western blot detecting Fun30-myc and alpha tubulin (loading control) in the indicated strains. To minimize proteolysis, we prepared whole cell lysates by a conventional TCA-extraction method.

2. The double mutant *exo1-nd fun30D* exhibits increased DSB, IS-dHJ and IH-dHJ relative to WT or any of the single mutants. Yet it fails to produce both inter-homolog CO and NCO (Fig. S4F), suggesting that the disappearance of the abundant IH DNA joint molecules between 8 and 12h mainly reflects a failure to repair off the homolog. Plotting these different data alongside each other would be useful.

We moved Figure S4E,F below **Figure 6E** as suggested. Two corrections to the reviewer's comments: a) IH-dHJs were not substantially increased in the double mutant compared to *exo1-nd* alone, and b) interhomolog crossovers and noncrossovers were greatly reduced in the double mutant, but there was not a complete failure to form them.

Furthermore, the CO-designated SEI intermediates are not different between *exo1-nd* and *exo1-nd fun30D* (Fig 6E), while CO production is. It suggests that, in a context in which resection is strongly limited, or in which *exo1* is catalytically inactive, Fun30 becomes critical for CO maturation. This point may be worth discussing.

Thank you for the comment. This is an interesting point, and we agree that a crossover maturation defect may be suggested by the much greater deficit in crossing over than in crossover-committed joint molecules. We added discussion of this point (P20, L453–459). We are not convinced that this makes a case for a specific role for Fun30 in this context, since a simple alternative is that the hypothetical crossover maturation defect is instead another consequence of the greatly shortened resection tracts.

3. Along this line, a refined analysis of SEI data presented in Figure 6 could yield greater insights into the mechanism by which Exo1 (and to a lesser extent Fun30) promotes homolog bias. Indeed, while the loss of the inter-homolog bias in the *fun30 exo1-nd* double mutant is very obvious at the dHJ level, this does not seem to be the case at the SEI level (gels Fig 6D). The SEI decomposition as IS and IH is not presented in Fig 6E, unlike for dHJ. If the defect in homolog bias indeed occurs at the SEI to dHJ transition, it may suggest a

defect in reuniting the two ends specifically on the homolog, ruling out defects in identifying the homolog, or in failing to suppress usage of the sister. Instead it would point at a defect in annealing the second end, which could be more easily overcome upon spatial tethering of the DSB and the donor (i.e. the sister). Such activity may require longer ssDNA, or be promoted by the role of Exo1 catalytic activity in DSB end-tethering in mitotic cells (Nakai .. Resnick DNA Repair 2011; Piazza .. Koszul NCB 2021), like other resection mutants. Although investigating the role of these proteins in DSB end-tethering in meiosis is out of the scope of this study, it could be mentioned in the relevant discussion section. In line with the interpretation that end-tethering may be compromised, both *exo1-nd* and *exo1-nd fun30* mutants exhibit reduced spore viability that cannot be explained solely by non-disjunction death. Instead, they could result from unrepaired, detached and randomly segregating chromosome fragments.

This is an interesting point, but we do not think the data support this interpretation. In the *exo1-nd fun30Δ* double mutant, noncrossover levels are reduced even more severely than crossovers. This would not be predicted if the defect in interhomolog bias occurs only after SEI formation specifically on the crossover pathway. Instead, the deficit in both crossovers and noncrossovers suggests a more global defect in interhomolog bias affecting all recombination pathways. Additionally, close inspection of the blots suggests that IS-SEI arcs are present in the double mutant; although we haven't performed experiments with Dad- or Mom-specific probes, such extended SEI arcs have been shown previously to indicate intersister bias (Kim et al, 2010). Since the same seems to be true here, this reinforces the interpretation that the defect in homolog bias occurs earlier, i.e., not at the SEI to dHJ transition. Because we do not feel that we have sufficient data to address this point clearly, we have elected not to add this to the paper.

4. A previous work from the team showed that S1-seq allows to detect ssDNA of opposite polarity to that expected for resection, that depended on Dmc1 (Mimitou 2017). This signal was interpreted as reflecting the presence of D-loop DNA joint molecules at the donor site. Analysis of this signal here may yield insights as to the individual and combined role of Fun30 and Exo1 catalytic function in D-loop formation and/or transition to intermediates detectable by 2D-gel (SEI, dHJs), and further substantiates the discussion about putative homology search defects.

We previously provided an analysis of these S1-seq signals in the *exo1-nd* mutant in the earlier study mentioned by the reviewer, showing that these putative recombination intermediates (RI) are positioned closer to hotspot centers. Not surprisingly, a similar change was observed in *fun30Δ* in keeping with the similar, but not identical, magnitude of resection defect (**Review Figure 2**). Because there is not much more we can glean from that observation alone, we elected not to put this in the paper.

The reviewer is undoubtedly more interested in the question of what happens in the *fun30Δ exo1-nd* double mutant. Unfortunately, however, the RIs become so short that they overlap with the resection tracts (**Review Figure 2**). This makes it impossible to quantify them. We can analyze these signals cleanly in wild type (or *exo1-nd* or *fun30Δ*) because the RIs appear to be long relative to the width of the hotspots themselves. Because resection tracts get so short in the double mutant, DSB distributions (i.e., hotspot widths) are now on the same size scale as the sequencing signal from the RIs, causing the overlap (see also Fig.

2A in the manuscript).

Review Figure 2. S1-seq reads from resection and recombination intermediates (RIs).

Hotspots were divided into quartiles based on Spo11-oligo counts in wild type (a proxy for DSB frequency), and the S1-seq profiles were averaged for each quartile after co-orienting top and bottom strand reads. The sequencing signal that appears to the left of hotspot centers is the wrong polarity to come from resection tracts, and in an *EXO1 FUN30* background this signal is Dmc1-dependent (Mimitou et al, 2017), so it is therefore thought to arise from recombination intermediates of unknown structure, possibly D-loops.

Minor comments:

- Genetic distance in Fig 5D would be more informative if data from the single *exo1-nd* and *fun30* mutants were shown alongside the double mutant.

We repeated the experiment including the single mutants. We now present the MI NDJ data in **Figure 5D** and the genetic distance data in **Appendix Table S2**. Unexpectedly, we found the genetic distances in the assayed intervals to be identical in *exo1-nd* and *exo1-nd fun30Δ*, with a reduction (relative to wild type) only in the most centromere-proximal interval (*CEN8-ARG4*). More puzzling, the *fun30Δ* single mutant had increased genetic distance in the *ARG4-THR1* interval. All of the differences were quantitatively small. We do not know the reason for the discrepancies, but we note that centromere-proximal regions have some unique recombination properties (Vincenten et al, 2015), so there may be a genomic position effect at play here. We consider the physical analysis to be more rigorous because of the number of replicates and the number of distinct loci assayed, so we presented the tetrad genetic distance data in **Appendix Tables S2, S3** and Methods for simplicity (P28, L617–630).

- Fig 6E: The sum of dHJ does not seem to correspond to the sum of IH- and IS-dHJ measured, particularly for the double mutant.

We are not sure why it appears this way to the reviewer. We checked all data plotted in **Figure 6E** but found no errors.

- The *exo1-nd* values are inconsistent between what is shown in plot in Fig 7C and 7D: IS/IH is ~1 in 7C and 0.25-0.5 in 7D.

Sorry for the confusion. **Figures 7D and 7E** were generated using datasets collected at the *HIS4LEU2* hotspot as described in the legend. **Figure 7C** is quantification of the data at *ERG1* (corresponding to the representative blots in **Figure 7B**). To clarify, we indicated the hotspot name in **Figures 7D and 7E**, and we adjusted the position of panel C to make its relation to panel B clearer.

- The S1-seq protocol appears to have been significantly improved relative to the previously published version. It may be of interest to the community to have a full updated protocol in supplementary materials in addition to the long list of modifications in the main methods. The detailed protocol is published (Mimitou & Keeney, 2018) and the modifications are all minor and can be easily cross-referenced to specific steps in the published version, so we have opted not to provide this.

Review References:

Byeon B, Wang W, Barski A, Ranallo RT, Bao K, Schones DE, Zhao K, Wu C, Wu WH (2013) The ATP-dependent chromatin remodeling enzyme Fun30 represses transcription by sliding promoter-proximal nucleosomes. *J Biol Chem* 288: 23182-23193

Kim KP, Weiner BM, Zhang L, Jordan A, Dekker J, Kleckner N (2010) Sister cohesion and structural axis components mediate homolog bias of meiotic recombination. *Cell* 143: 924-937

Mimitou EP, Keeney S (2018) S1-seq Assay for Mapping Processed DNA Ends. *Methods Enzymol* 601: 309-330

Mimitou EP, Yamada S, Keeney S (2017) A global view of meiotic double-strand break end resection. *Science* 355: 40-45

Neves-Costa A, Will WR, Vetter AT, Miller JR, Varga-Weisz P (2009) The SNF2-family member Fun30 promotes gene silencing in heterochromatic loci. *PLoS One* 4: e8111

Pan J, Sasaki M, Kniewel R, Murakami H, Blitzblau HG, Tischfield SE, Zhu X, Neale MJ, Jasin M, Socci ND *et al* (2011) A hierarchical combination of factors shapes the genome-wide topography of yeast meiotic recombination initiation. *Cell* 144: 719-731

Vincenten N, Kuhl LM, Lam I, Oke A, Kerr AR, Hochwagen A, Fung J, Keeney S, Vader G, Marston AL (2015) The kinetochore prevents centromere-proximal crossover recombination during meiosis. *Elife* 4

Zakharyevich K, Ma Y, Tang S, Hwang PY, Boiteux S, Hunter N (2010) Temporally and biochemically distinct activities of Exo1 during meiosis: double-strand break resection and resolution of double Holliday junctions. *Mol Cell* 40: 1001-1015

Dr. Scott Keeney
Howard Hughes Medical Institute, Memorial Sloan-Kettering Cancer Center
Molecular Biology Program
1275 York Ave
Box 97
New York, NY 10065

12th Nov 2024

Re: EMBOJ-2024-117736R
Meiotic DNA break resection and recombination rely on chromatin remodeler Fun30

Dear Dr. Keeney,

Thank you for submitting your final revised manuscript for our consideration. I am pleased to inform you that we have now accepted it for publication in The EMBO Journal.

Yours sincerely,

Hartmut Vodermaier

Referee #2:

The authors carefully responded to all of my concerns. It's a really nice story.

Referee #3:

The authors provided extensive and satisfactory answers to my comments. Although inconclusive for the purpose of this work, the Fun30-K603R data could be provided as a supplementary figure, so that any other lab interested in the matter may be aware of the effect of this mutation on protein levels.
